# On the Suboptimality of Thompson Sampling in High Dimensions

**Raymond Zhang**
ENS Paris Saclay, Département de Mathématiques, Gif-sur-Yvette, France
`raymond.zhang@ens-paris-saclay.fr`

**Richard Combes**
Centrale-Supelec, Laboratoire des Signaux et Systèmes (L2S), Gif-sur-Yvette, France
`richard.combes@centralesupelec.fr`

## Abstract

In this paper we consider Thompson Sampling (TS) for combinatorial semi-bandits. We demonstrate that, perhaps surprisingly, TS is sub-optimal for this problem in the sense that its regret scales exponentially in the ambient dimension, and its minimax regret scales almost linearly. This phenomenon occurs under a wide variety of assumptions including both non-linear and linear reward functions, with Bernoulli distributed rewards and uniform priors. We also show that including a fixed amount of forced exploration to TS does not alleviate the problem. We complement our theoretical results with numerical results and show that in practice TS indeed can perform very poorly in some high dimensional situations.

## 1 Introduction

We consider the problem of combinatorial bandits with semi-bandit feedback. At time $t = 1, , ..., T$ a learner selects a decision $x(t) \in \mathcal{X}$ where $\mathcal{X} \subset \{0, 1\}^d$ is the set of available decisions. The environment then draws a random vector $Z(t) \in \mathbb{R}^d$. The learner then observes $Y(t) = x(t) \odot Z(t)$, where $\odot$ denotes the Hadamard (elementwise) product. This setting is called semi bandit feedback. We assume that $(Z(t))_{t \geq 1}$ are i.i.d., and that $Z_1(t), ..., Z_d(t)$ are independent and distributed as $Z_i(t) \sim \text{Bernoulli}(\theta_i)$ for all $t,i$. Then the learner receives a reward $f(x(t), Z(t))$ where $f$ is a known function.

The goal is to minimize the regret:

$$R(T, \theta) = T \max_{x \in \mathcal{X}} \left\{ \mathbb{E} f(x, Z(t)) \right\} - \sum_{t=1}^{T} \mathbb{E} f(x(t), Z(t)).$$

Initially $\theta$ is unknown to the learner and minimizing regret involves exploring suboptimal decisions just enough in order to identify the optimal decision. For any decision $x \in \mathcal{X}$, define the reward gap

$$\Delta_x = \max_{x \in \mathcal{X}} \{ \mathbb{E} f(x, Z(t)) \} - \mathbb{E} f(x, Z(t)),$$

which is the amount of regret incurred by choosing $x$ instead of an optimal decision

$$x^\star \in \arg \max_{x \in \mathcal{X}} \left\{ \mathbb{E} f(x, Z(t)) \right\},$$

and $\Delta_{\min} = \min_{x \in \mathcal{X} : \Delta_x > 0} \Delta_x$ the minimal gap. We define $m \triangleq \max_{x \in \mathcal{X}} \sum_{i=1}^{d} |x_i|$ the size of the maximal decision.

35th Conference on Neural Information Processing Systems (NeurIPS 2021).

For this problem, an algorithm which has attracted a lot of interest is Thompson Sampling (TS), which at time $t$ selects the decision maximizing $x \mapsto f(x, V(t))$ where $V(t)$ is a random variable distributed as the posterior distribution of $\theta$ knowing the information available at time $t$, which is $Y(1), ..., Y(t-1)$. The prior distribution of $\theta$ can be chosen in various ways, the most natural being a non-informative distribution such as the uniform distribution.

TS is usually computationally simple to implement, for instance when $f$ is linear, since it involves maximizing $f$ over $\mathcal{X}$. Also, for some problem instances it tends to perform well numerically. A particular case of interest is *linear combinatorial semi-bandits* where $f(x, \theta) = \theta^\top x$ so that the reward is a linear function of the decision.

**Our contribution.** We show that the regret of TS in general does not scale polynomially in the ambient dimension $d$.

(i) We provide several examples, both for linear and non-linear combinatorial bandits, where the regret of TS does not scale polynomially in the dimension $d$ (in fact in some cases it may scale even faster than exponentially in the dimension). In some cases, we show that one must wait for an amount of time greater than $\Omega(d^d)$ for TS to perform at least as well as random choice where one simply chooses $x(t)$ uniformly distributed in $\mathcal{X}$ at every round. Therefore, in high dimensions, in some instances, TS in general can perform strictly worse than random choice for all practically relevant time horizons.

(ii) We show that the minimax regret of TS scales at least as $\Omega(T^{1-\frac{1}{d}})$ so that it is not minimax optimal, as there exists algorithms such as CUCB and ESCB with minimax regret $O(\mathbf{poly}(d)\sqrt{T(\ln T)})$. In fact, in high dimensions, the minimax regret of TS is almost linear.

(iii) We further show that adding forced exploration as an initialization step to TS does not correct the minimax problem, so that this is not an artifact due to initialization.

(iv) Using numerical experiments, we show that indeed, for reasonable time horizons, TS performs very poorly in high dimensions in some instances.

We believe that our results highlight two general characteristics of TS. First, TS tends to be much more greedy than optimistic algorithms such as ESCB and CUCB. This greedy behavior explains why the regret of TS is, in some instances, much smaller than that of optimistic algorithms. In fact it is sometimes so greedy that it misses the optimal decision. Second, TS tends to be by nature a "risky" algorithm so that its regret exhibits very large fluctuations across runs. In some cases it finds the optimal arm very quickly and with little to no regret, while in other cases it simply misses the optimal decision and performs worse than random choice.

**Related work.** Combinatorial bandits are a generalization of classical bandits studied in [15]. Several asymptotically optimal algorithms are known for classical bandits, including the algorithm of [16], KL-UCB [4], DMED [12] and TS [20, 13]. Other algorithms include the celebrated UCB1 [2]. A large number of algorithms for combinatorial semi-bandits have been proposed, many of which naturally extend algorithms for classical bandits to the combinatorial setting. CUCB [5, 14] is a natural extension of UCB1 to the combinatorial setting. ESCB [7, 9] is an improvement of CUCB which leverages the independence of rewards between items. AESCB [8] is an approximate version of ESCB with roughly the same performance guarantees and reduced computational complexity. TS for combinatorial bandits was considered in [10, 21, 18]. Also, combinatorial semi bandits are a particular case of structured bandits, for which there exists asymptotically optimal algorithms such as OSSB [6]. Table 1 presents the best known regret upper bounds for CUCB, ESCB and TS. For completeness, we also recall the complete regret upper bound for ESCB as Theorem 9 (Appendix B).

We provide two types of bounds for TS: problem dependent bounds (sometimes called gap-dependent bounds) and minimax bounds (or gap-free bounds). The former involves $T$, $d$, $m$ and $\Delta_{\min}$, while the latter hold for any value of $\Delta_{\min}$.

An important observation is that all of the known regret upper bounds for TS [10], [21], [18] feature at least one term that does not scale polynomially with the dimension. In particular, the paper [18] shows that there exists a universal constant $C \geq 0$ such that the regret of TS is upper bounded by

$$R(T) \leq C \Big[ d(\ln m)^2 \ln(|\mathcal{X}|T)/\Delta_{\min} + dm^3/\Delta_{\min}^2 + m((m^2+1)/\Delta_{min})^{2+4m} \Big].$$

This is a general bound for all combinatorial sets of interest. The bound has a super exponential term (that does not depend on $T$) : $m((m^2+1)/\Delta_{min})^{2+4m}$. Regret bounds for CUCB and ESCB do

Table 1: Algorithms and best known regret bounds.

| Algo. | Regret |
|---|---|
| | ((i) problem dependent and (ii) minimax) |
| CUCB | (i) $O(dm(\ln T)/\Delta_{min})$ [14][Theorem 4] |
| | (ii) $O(\sqrt{dmT(\ln T)} + dm)$ [14][Theorem 6] |
| ESCB | (i) $O(d(\ln m)^2(\ln T)/\Delta_{min} + dm^3/\Delta_{min}^2)$ [9][Theorem 2] |
| | (ii) $O(\sqrt{d(\ln m)^2 T(\ln T)} + dm)$ [9][Corollary 1] |
| TS | (i) $O(d(\ln m)^2 \ln(|\mathcal{X}|T)/\Delta_{min} + dm^3/\Delta_{min}^2 + m((m^2+1)/\Delta_{min})^{2+4m})$ |
| | [18][Theorem 1] |
| | (ii) not available |

not feature this exponential dependency in the dimension. However, since TS tends to perform very well in all of the numerical experiments presented in the literature, a natural intuition would be that all known upper bounds are simply not sharp, and that the true regret of TS does not really grow exponentially with the dimension. We show in this work that this intuition is incorrect, and that the regret of TS really does scale (at least) exponentially in the dimension i.e. it suffers from the "curse of dimensionality". This directly implies that, in high dimensions, and for some combinatorial sets, CUCB and ESCB perform much better than TS.

In order to alleviate this problem, it would be natural to attempt to modify the prior distribution used by TS, since it is known to have a strong influence on its performance [3, 17]. Similarly, in a related problem, [1] suggests to use correlated Thompson samples, and [11] studies the influence of the prior on numerical performance. We believe that this is an interesting open problem.

---

Prior knowledge to the learner: combinatorial set $\mathcal{X} \subset \{0,1\}^d$, function $f : \mathcal{X} \times [0,1]^d \to \mathbb{R}$
For $t = 1, ..., T$:
    1. The learner computes statistics $A(t) = \sum_{s=1}^{t-1} Z(s) \odot x(s)$ and $B(t) = \sum_{s=1}^{t-1} x(s) - Z(s) \odot x(s)$
    2. The learner draws $V(t)$ with $V_1(t), ..., V_d(t)$ independent and $V_i(t) \sim \text{Beta}(A_i(t) + 1, B_i(t) + 1)$
    3. The learner chooses decision $x(t) \in \arg\max_{x \in \mathcal{X}}\{f(x, V(t))\}$
    4. The environment draws $Z(t)$ with $Z_1(t), ..., Z_d(t)$ independent and $Z_i(t) \sim \text{Ber}(\theta_i)$
    5. The learner observes $Z(t) \odot x(t)$ and receives the reward $f(x(t), Z(t))$
Performance metric: expected regret $R(T, \theta) = T(\max_{x \in \mathcal{X}}\{\mathbb{E}f(x, Z(t))\}) - \sum_{t=1}^{T} \mathbb{E}f(x(t), Z(t))$

Figure 1: TS for combinatorial semi-bandits with Bernoulli rewards and uniform prior.

## 2 Model

### 2.1 Problem Dependent Regret and Minimax Regret

In order to evaluate the performance of an algorithm over the set of instances $\theta \in [0,1]^d$, there are two main figures of merit that we study in this paper. The first is the problem-dependent regret which is $R(T, \theta)$ when $\theta \in [0,1]^d$ is fixed. The second is the minimax regret which is the worse case over $\theta$ for $T$ fixed: $\max_{\theta \in [0,1]^d} R(T, \theta)$.

### 2.2 TS

The basic TS algorithm works as follows. For $i = 1, ..., d$, define $A_i(t) = \sum_{s=1}^{t-1} Z_i(s) x_i(s)$ and $B_i(t) = \sum_{s=1}^{t-1} (1 - Z_i(s)) x_i(s)$ which represent the number of successes and failures observed when getting a sample to estimate $\theta_i$. We define $N_i(t) = A_i(t) + B_i(t) = \sum_{s=1}^{t-1} x_i(s)$ the number of samples available at time $t$ to estimate $\theta_i$, and

$$\hat{\theta}_i(t) = \frac{A_i(t)}{\max(N_i(t), 1)},$$

the corresponding estimate of $\theta_i$ which is simply the empirical mean.

The TS algorithm selects decision

$$x(t) \in \arg\max_{x \in \mathcal{X}} f(x, V(t)) \text{ where } V_i(t) \sim \text{Beta}(A_i(t) + 1, B_i(t) + 1),$$

and $V_1(t), ..., V_d(t)$ are independent. Vector $V(t)$ is called the *Thompson sample* at time $t$. TS is based on a Bayesian argument, $V(t)$ is drawn according to the posterior distribution of $\theta$ knowing the information available at time $t$, where the prior distribution for $\theta$ is uniform over $[0, 1]^d$. Choosing decision $x(t)$ as done above should ensure that one explores just enough to find the optimal decision. In the linear case, the decision can be computed by linear maximization over $\mathcal{X}$ :

$$x(t) \in \arg\max_{x \in \mathcal{X}} \{V(t)^\top x\}.$$

This explains the practical appeal of TS, since whenever linear maximization over $\mathcal{X}$ can be implemented efficiently, the algorithm has low computational complexity.

### 2.3   TS with Forced Exploration

A natural extension of TS is to add $\ell$ forced exploration rounds, where $\ell$ is a fixed number, in order to avoid some artifacts that could possibly occur due to the prior distribution. The algorithm operates as follows. At time $1 \le t \le \ell$, one selects $x(t) \in \mathcal{X}$ such that $x_{i(t)}(t) = 1$ with

$$i(t) \in \min_{i=1,...,d} N_i(t).$$

Otherwise for $t \ge \ell + 1$ one selects

$$x(t) \in \arg\max_{x \in \mathcal{X}} f(x, V(t)),$$

with $V(t)$ the Thompson sample defined above. Namely, one first performs a forced exploration during $\ell$ rounds then apply TS. This guarantees that $N_i(t) \ge \lfloor \ell/d \rfloor$ samples are available to estimate $\theta_i$ for all $i = 1, ..., d$, then one subsequently applies TS. We call this variant TS with $\ell$ forced exploration rounds.

## 3   Main Results

We now state our main theoretical results. All proofs are found in the appendix.

### 3.1   Some Combinatorial Sets of Interest

We will provide several examples of combinatorial sets where the regret of TS indeed scales exponentially with the dimension, so that this phenomenon is quite general and is not an artifact that only occurs for one particular family of combinatorial structures. We define $\mathcal{X}^p$ the set of paths of the directed acyclic graph depicted in figure 2. This set has two disjoint decisions $(1, ..., 1, 0, ..., 0)$ and $(0, ..., 0, 1, ..., 1)$ of equal size $m = \frac{d}{2}$. We define $\mathcal{X}^m$ the set of matchings of the bipartite graph depicted in figure 2. This graph has $d$ vertices and $d$ edges.

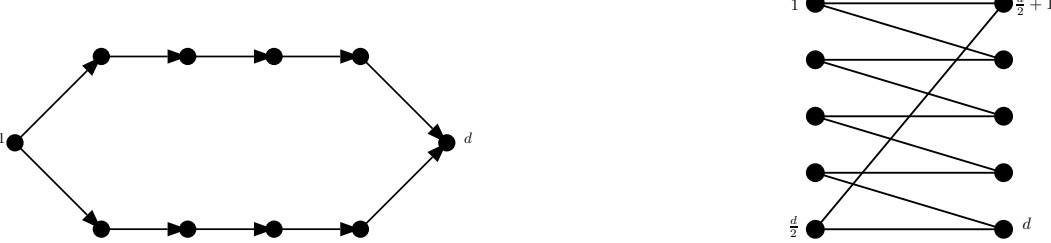

Figure 2: Paths in a directed acyclic graph (left) and matchings of the Z graph (right)

The combinatorial sets presented are very simple. However our results can by generalised for more complex set of interest without losing the exponential nature of the regret. For example the two path

environment can be generalized to $k > 2$ paths. It can also be generalized for non disjoint paths if the optimal path does not share "a lot" of edges with all the other paths. This could be the case for real life applications like shortest path routing or in medical trials where treatments cannot be associated with each other. With those simple examples in mind many other more complex sets that exhibit exponential regret can be found. However we do not provide formal proof for those more complex examples as the simple example of paths is sufficient to prove the suboptimality of TS here.

## 3.2 Linear Combinatorial Bandits

We focus on linear bandits, where the expected reward function is linear i.e. $f(x, \theta) = x^\top \theta$. For the non linear case see Appendix C. There, we consider non linear problems inspired by [21][Theorem 3], where the Thompson sample of sub-optimal decisions has no variance. One could be lead to think that the exponential dependency of the regret on the dimension could be caused by this feature, and it is hence natural to investigate the linear case, which is not only more common, but also where the Thompson sample of any decision always has a non-null variance.

In Theorem 1 we consider a linear problem over the combinatorial set $\mathcal{X}^p$ which is formed of two disjoint paths. We show that the regret of TS does scale exponentially in the dimension for this problem. Therefore this phenomenon is not linked to a particular, well chosen, non-linear reward function, but also occurs for the classical case of linear reward functions.

**Theorem 1.** *Consider a linear combinatorial bandit problem over combinatorial set $\mathcal{X}^p$ and parameter $\theta_i = 1$ if $1 \leq i \leq d/2$ and $\theta_i = 1 - \frac{\Delta}{m}$ otherwise. Assume that $\frac{\Delta}{m} + \frac{1}{\sqrt{m}} < \frac{1}{2}$.*

*Then the regret of TS is lower bounded by*

$$R(T, \theta) \geq \frac{\Delta}{4p_\Delta}(1 - (1 - p_\Delta)^{T-1}), \text{ with } p_\Delta = \exp\left\{-\frac{2m}{9}\left(\frac{1}{2} - (\frac{\Delta}{m} + \frac{1}{\sqrt{m}})\right)^2\right\}.$$

Theorem 1 is proven by showing that the first time that the optimal decision is selected is exponentially large in general. The central argument can be summarized as follows. Consider $t$ such that at times $1, ..., t$ only the suboptimal decision has been selected. The probability of selecting the optimal decision is $\mathbb{P}\left(\sum_{i=1}^m V_i(t) \geq \sum_{i=m+1}^d V_i(t) | A(t), B(t)\right)$ where $V_1(t), ..., V_d(t)$ are independent, distributed in $[0, 1]$, and their respective expectations are

$$\mathbb{E}\left(V_i(t) | A(t), B(t)\right) = \begin{cases} \frac{1}{2}, & \text{if } 1 \leq i \leq m, \\ \frac{A_i(t)+1}{t+2}, & \text{if } m+1 \leq i \leq d. \end{cases}$$

Furthermore, from the law of large numbers, when $t$ is large, $\sum_{i=m+1}^d A_i(t) \approx (1 - \frac{\Delta}{m})t$ since we have sampled the sub-optimal decision $t$ times. Therefore $\sum_{i=1}^m \mathbb{E}(V_i(t)|A(t), B(t)) = \frac{m}{2}$ and again because t is large, $\sum_{i=m+1}^d \mathbb{E}(V_i(t)|A(t), B(t)) \approx m - \Delta$. Since $V_1(t), ..., V_d(t)$ are independent and distributed in $[0, 1]$, their sums must concentrate around their expectation, and from Hoeffding's inequality:

$$\mathbb{P}\left(\sum_{i=1}^m V_i(t) \geq \sum_{i=m+1}^d V_i(t) | A(t), B(t)\right) \leq O\left(e^{-um((\frac{1}{2} - \frac{\Delta}{m}))^2}\right),$$

where $u > 0$ is some positive exponent related to how concentrated the Thompson samples are. This implies that, for large $t$, the probability of selecting the optimal decision is exponentially small if it has never been selected previously.

Also, we see that this phenomenon of lack of exploration by TS is a typically high dimensional phenomenon. In short, the Thompson samples of decisions will tend to concentrate around their expectation, so that TS will, most of the time, act greedily and simply select the decision maximizing the empirical reward. We can also emphasize the fact that when $m$ grows, we can have an arbitrary large gap $\Delta$ and still have exponential regret. This is unexpected, since the difficulty of a bandit problem is usually a decreasing function of the gap $\Delta$. Furthermore another version of Theorem 1 which exhibits exponential behavior can be shown with parameters $\theta_i = u$ if $1 \leq i \leq d/2$ and $\theta_i = u - \frac{\Delta}{m}$ otherwise, under the condition $\frac{\Delta}{m} + \frac{1}{\sqrt{m}} < u - \frac{1}{2}$ where $u \in ]\frac{1}{2}, 1]$. We chose the parameters of Theorem 1 for the sake of clarity and being at the edge of the parameter space is not a necessary condition to have exponential regret.

## 3.3 Linear Combinatorial Bandits: Small Gap Regime

Theorem 2 is another regret bound for TS which is more accurate in the regime where $\Delta$ is small, it allows to deduce a lower bound for the minimax regret as well. The proof is a more intricate version of the proof of Theorem 1 highlighted above. Corollary 3 states that the minimax regret of TS scales as $O(T)$ for $T \leq m!$, and at least as $\Omega(T^{1-\frac{2}{d}})$ for $T \geq m!$. Of course, in practice, when $m$ is large we have $T \leq m!$ for any reasonable time horizon, so that the regret of TS is linear in this regime. Also, as stated by Corollary 3, TS is provably not minimax optimal, as there exist algorithms with minimax regret scaling at most as $O(d^{1/4}\sqrt{T \ln T})$.

**Theorem 2.** *Consider a linear combinatorial bandit problem over combinatorial set $\mathcal{X}^p$ and parameter $\theta_i = 1$ if $1 \leq i \leq d/2$ and $\theta_i = 1 - \frac{\Delta}{m}$ otherwise.*

*Then there exists universal constants $C_2, C_3$ such that for all $T \geq T_0(m) \equiv C_2 m^2 \ln m$, and $\Delta \leq 1/6$ and $m \geq 5$, the regret of TS is lower bounded by*

$$R(T,\theta) \geq \frac{\Delta}{4p_\Delta}(1 - (1 - p_\Delta)^{T_0 - 1}) + C_3 \Delta \frac{(1 - g_\Delta)^{T_0} - (1 - g_\Delta)^{T - T_0 + 1}}{g_\Delta}$$

*with*

$$g_\Delta = \frac{(2\Delta)^m}{m!} \text{ and } p_\Delta = \exp\left\{-\frac{2m}{9}\left[\frac{1}{2} - \left(\frac{\Delta}{m} + \frac{1}{\sqrt{m}}\right)\right]^2\right\}$$

**Corollary 3.** *Consider a linear combinatorial bandit problem over combinatorial set $\mathcal{X}^p$ with $m \geq 5$. If $T > 2^m m!$ then the minimax regret of TS is lower bounded by*

$$\max_{\theta \in [0,1]^d} R(T,\theta) \geq C_4 m T^{1-\frac{2}{d}}$$

*Otherwise it is lower bounded by*

$$\max_{\theta \in [0,1]^d} R(T,\theta) \geq C_4' T$$

*with $C_4, C_4' > 0$ universal constants.*

*The minimax regret of ESCB for this set $\mathcal{X}^p$ is upper bounded by:*

$$\max_{\theta \in [0,1]^d} R(T,\theta) \leq C_5 d^{\frac{1}{4}}\sqrt{T \ln T}.$$

*with $C_5 \geq 0$ a universal constant. Therefore TS is* not *minimax optimal.*

## 3.4 Linear Combinatorial Bandits with Forced Exploration

We finally extend our results to show that, even when forced exploration is added, TS still provably incurs a regret growing exponentially with the dimension, as stated by Theorem 4 and Theorem 5. In particular, if the number of forced exploration rounds $\ell$ satisfies

$$\ln\left(\frac{1}{1 - \frac{\Delta}{m}}\right) \leq \frac{2\left(\frac{1}{\frac{\ell}{2}+2} - \left(\frac{\Delta}{m} + \frac{1}{\sqrt{m}}\right)\right)^2}{(\frac{\ell}{2} + 3)^2 \frac{\ell}{2}}$$

then Theorem 4 implies that the regret of TS still increases exponentially in $m$, in spite of the forced exploration added to the algorithm. In fact it is impossible to set $\ell$ to prevent exponential regret from happening, unless the learner knows the value of the gap $\Delta$ in advance. Indeed, for any fixed $\ell$, the above inequality always holds providing that $\Delta$ is small enough.

**Theorem 4.** *Consider a linear combinatorial bandit problem over combinatorial set $\mathcal{X}^p$ and parameter $\theta_i = 1$ if $1 \leq i \leq d/2$ and $\theta_i = 1 - \frac{\Delta}{m}$ otherwise.*

*Then if $\frac{\Delta}{m} + \frac{1}{\sqrt{m}} < \frac{1}{\frac{\ell}{2}+2}$ the regret of TS with $\ell$ forced exploration rounds is lower bounded by*

$$R(T,\theta) \geq (1 - \frac{\Delta}{m})^{\frac{m\ell}{2}} \frac{\Delta}{4p_\Delta^\ell}(1 - (1 - p_\Delta^\ell)^{T-1}), \text{ with}$$

$$p_\Delta^\ell = \exp\left\{-2m\left(\frac{1}{\frac{\ell}{2}+2} - \left(\frac{\Delta}{m} + \frac{1}{\sqrt{m}}\right)\right)^2 / (\frac{\ell}{2} + 3)^2\right\}.$$

**Theorem 5.** *Consider a linear combinatorial bandit problem over combinatorial set $\mathcal{X}^p$ with $m \geq 5$ and parameter $\theta_i = 1$ if $1 \leq i \leq d/2$ and $\theta_i = 1 - \frac{\Delta}{m}$ otherwise.*

*Then if $T > 4^m m!$ and $T > T_0(m,l) \equiv C_6 m^2 (\ln m) \ell^{\frac{1}{4} - \frac{1}{m}}$ the minimax regret of TS with $\ell$ forced exploration rounds is lower bounded by*

$$\max_{\theta \in [0,1]^d} R(T, \theta) \geq C_7 C(\ell, m) \frac{m}{\ell} T^{1 - \frac{1}{m}}.$$

*Otherwise it is bounded by*

$$\max_{\theta \in [0,1]^d} R(T, \theta) \geq C_7' C(\ell, m) \frac{T}{\ell}.$$

*with $C_6, C_7, C_7' > 0$ universal constants and $C$ such that $\forall \ell \in \mathbb{N}, \lim_{m \to \infty} C(\ell, m) = 1$*

## 4  Numerical Experiments

We now illustrate the exponential regret of TS in practical settings using numerical experiments. Due to this exponential nature, some of those experiments involve a significant amount of computing time in high dimensions. Due to limited space, we solely consider the linear case, which is the most often considered in the literature. Unless specified otherwise we use 1000 independent sample paths for averaging, and 95% confidence intervals are presented on the plots.

**First selection of the optimal decision**    As shown by our theoretical results, the first time that the optimal decision is selected $\tau = \min\{t \geq 1 : x(t) = x^\star\}$ can be exponentially large, and this is what causes exponential regret. On Figure 3, we present the c.d.f. (cumulative distribution function) of $\tau$ as a function of $m$ for combinatorial sets $\mathcal{X}^p$ and $\mathcal{X}^m$ introduced above. The parameter values are chosen as in the previous sections $\theta_i = 1$ if $1 \leq i \leq d/2$ and $\theta_i = \frac{\Delta}{m}$ otherwise. For each sample path we generate $\tau$ by simulating TS until the optimal decision is played for the first time.

Some quantiles of $\tau$ indeed seem to increase exponentially as $m$ grows and for reasonable values of $m$, $\tau$ can be very large with appreciable probability, for instance on Fig. 3 for $m = 14$, $\tau \geq 5.10^4$ with probability greater than $0.1$. Clearly, on sample paths where this happens, TS performs worse than random choice for the first $5.10^4$ time steps which is a surprisingly poor behaviour, especially on such a simple problem. This also showcases the fact that those sample paths happen relatively often. Thus the regret of TS is not only due to very rare occasions with high regret but also because of those poor behavior that can happen quite often.

To investigate the impact of the gap $\Delta$, on Figure 4a and Figure 4b, we plot the expected first time the optimal decision is selected $\mathbb{E}(\tau)$ as a function of $m$ for various $\delta = \frac{\Delta}{m}$, for the sets of paths $\mathcal{X}^p$ and matchings $\mathcal{X}^m$. Once again we observe an exponential growth in both figures, and this growth is particularly fast for small values of $\delta$. When the gap gets small, the exponential growth of regret is exacerbated, leading to an even worse performance.

**Impact of forced exploration**    We now investigate if forced exploration alleviates the problem in practice, and consider $\ell$ forced exploration rounds. On figures 5a and 5b we plot the expected first time the optimal decision is selected $\mathbb{E}(\tau)$ as a function of $m$ for various $\delta = \frac{\Delta}{m}$ and $\ell$, for the sets of paths $\mathcal{X}^p$. As predicted by Theorem 4, $\mathbb{E}(\tau)$ still seems to scale exponentially in $m$ which causes exponential regret.

Theorem 4 states that if $\ell$ is chosen such that $\frac{\Delta}{m} < \frac{2}{\ell + 1}$ then regret scales exponentially. On the other hand one could think that when $\ell$ is chosen large enough to violate this condition then regret does not grow as rapidly. Figure 5b shows that, at least numerically, this does not appear to be the case, indeed, we choose $\ell = m$, and there are values of $\delta$ such that the regret still seem to scale exponentially in $m$.

**Comparison with optimistic algorithms**    We now compare TS with the state-of-the-art frequentist algorithms ESCB and/or CUCB. These experiments are averaged over 40 sample paths due to computational limits. On figures 6a and 6b we present the regret as a function of $m$ for the set of paths $\mathcal{X}^p$, $\frac{\Delta}{m} = 0.1$, $T = 4.10^4$ and the set of matchings $\mathcal{X}^m$, $\frac{\Delta}{m} = 0.05$, $T = 4.10^4$ respectively. The results show that the regret of TS is larger than that of CUCB and/or ESCB by several orders of magnitude in high dimensions, as predicted by our theoretical results. In fact the regret of TS is

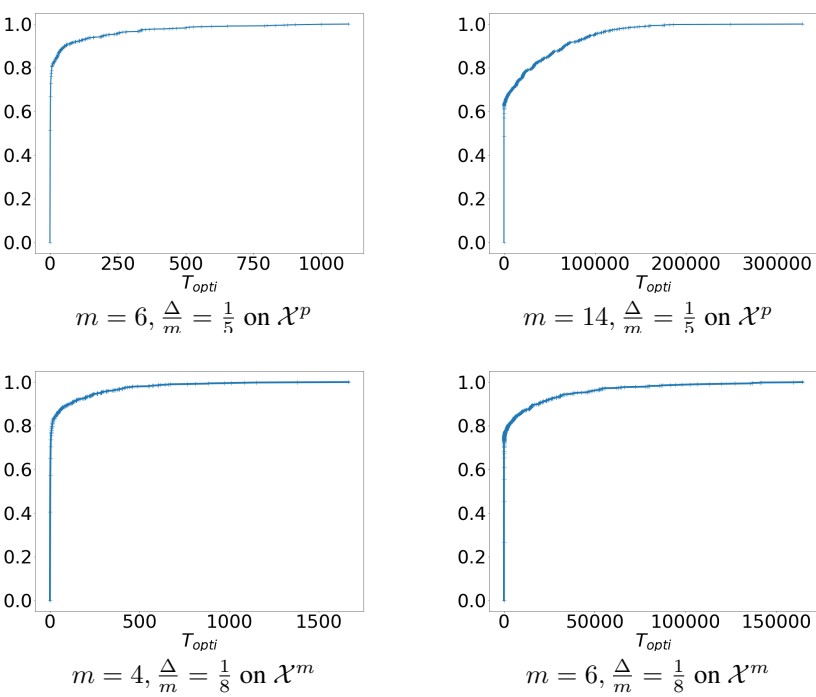

Figure 3: C.d.f. of the first time the optimal decision is played $\tau$ as a function of $m$ for set of paths and matchings $\mathcal{X}^p$, $\mathcal{X}^m$

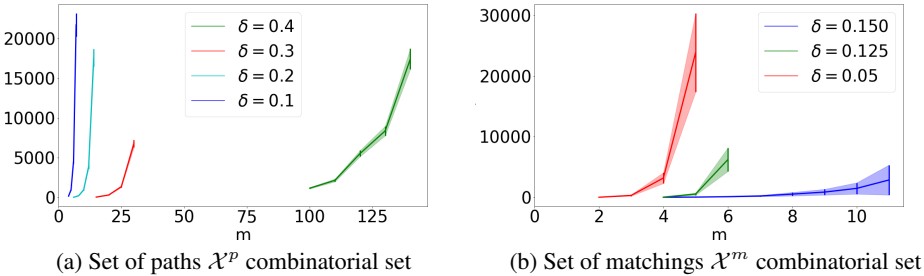

(a) Set of paths $\mathcal{X}^p$ combinatorial set

(b) Set of matchings $\mathcal{X}^m$ combinatorial set

Figure 4: Expectation of the first time the optimal decision is played $\mathbb{E}(\tau)$ as a function of $m$ and various values of $\frac{\Delta}{m} = \delta$

so overwhelmingly large that, due to the scale of the figure, it looks like the regret of ESCB and/or CUCB does not increase with the dimension (this is of course not the case). On figures 7a and 7b we perform similar experiments but with smaller gaps, $\frac{\Delta}{m} = \frac{1}{m}$ and the same behaviour arises.

## 5   Conclusion

We have shown through both theoretical analysis as well as numerical experiments that TS can perform very poorly in high dimensions, both for both linear and non linear problems, and for various combinatorial structures such as sets of paths and matchings (one could consider more complex combinatorial set including multiple non disjoint paths). Introducing forced exploration does not alleviate the problem either. Therefore, this is not an artifact, but rather a general problem. In essence, Thompson performs poorly because it has a tendency to play much too greedily, and in high dimensions this sometimes leads to a complete lack of exploration and missing the optimal arm. Our work points towards a new challenging open problem which is to design better TS-like

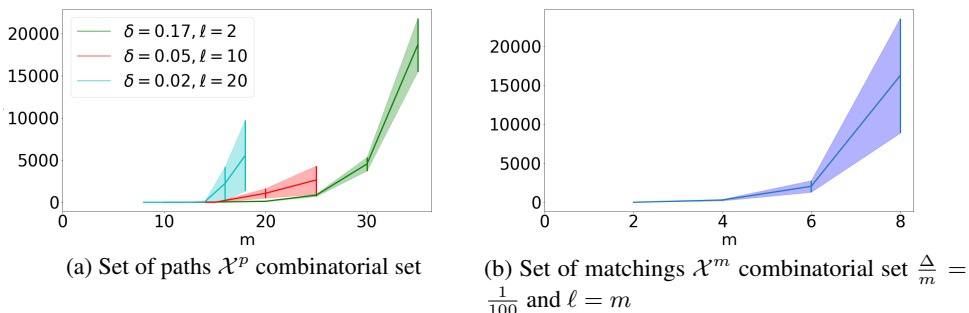

(a) Set of paths $\mathcal{X}^p$ combinatorial set

(b) Set of matchings $\mathcal{X}^m$ combinatorial set $\frac{\Delta}{m} = \frac{1}{100}$ and $\ell = m$

Figure 5: Expectation of the first time the optimal decision is played $\mathbb{E}(\tau)$ as a function of $m$ and various values of $\frac{\Delta}{m} = \delta$ and $\ell$

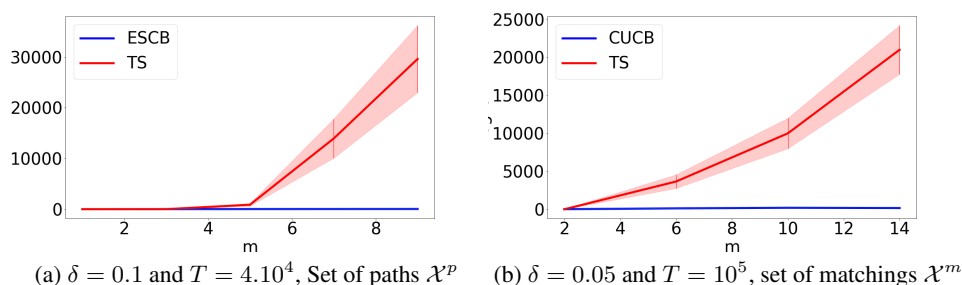

(a) $\delta = 0.1$ and $T = 4.10^4$, Set of paths $\mathcal{X}^p$

(b) $\delta = 0.05$ and $T = 10^5$, set of matchings $\mathcal{X}^m$

Figure 6: Regret comparison between ESCB and TS for set of paths $\mathcal{X}^p$ and matching $\mathcal{X}^m$ (Averaged over 40 experiences)

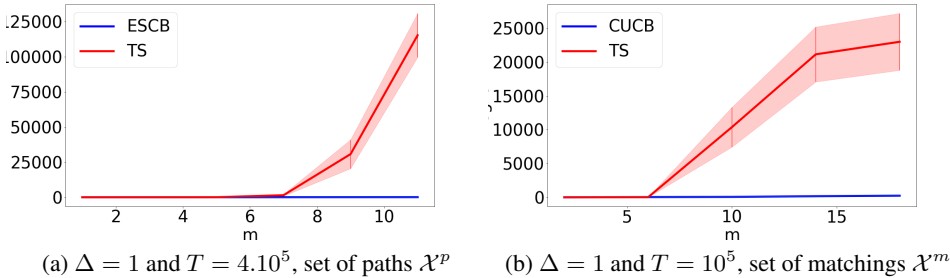

(a) $\Delta = 1$ and $T = 4.10^5$, set of paths $\mathcal{X}^p$

(b) $\Delta = 1$ and $T = 10^5$, set of matchings $\mathcal{X}^m$

Figure 7: Regret comparison between ESCB and TS for set of paths $\mathcal{X}^p$ and matching $\mathcal{X}^m$ For a fixed $\Delta$ (Averaged over 20 and 40 experiences)

algorithms for regret minimization that can deal with high-dimensional problems, while retaining the computational efficiency of TS. Two reasonable ideas to explore would be (i) carefully designing the prior distribution (ii) enforcing forced explorations at regular intervals, possibly in an adaptive manner. Also, our work concerns the Bernoulli setting but we believe that our results can be generalized to bounded distributions. It is not obvious whether or not our results still hold for Gaussian distributions, and seems like an interesting open problem.

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
