# A Proofs

## A.1 Technical Results

We state a technical result about the product of i.i.d. random variables with Beta distribution.

**Lemma 6.** *Let $V_1, ..., V_m$ i.i.d. with distribution $V_i \sim Beta(\alpha, 1)$. Then for all $\Delta \in [0, 1]$:*

$$\mathbb{P}\left(\prod_{i=1}^{m} V_i \geq 1 - \Delta\right) \leq \frac{\alpha^m}{m(m!)} \left[\ln\left(\frac{1}{1 - \Delta}\right)\right]^m.$$

*Proof.* Taking logarithms:

$$\mathbb{P}\left(\prod_{i=1}^{m} V_i \geq 1 - \Delta\right) = \mathbb{P}\left(\sum_{i=1}^{m} \ln \frac{1}{V_i} \leq \ln\left(\frac{1}{1 - \Delta}\right)\right)$$

Now if $V_i \sim \text{Beta}(\alpha, 1)$ then $\ln \frac{1}{V_i} \sim \text{Exp}(\alpha)$ and since $V_i$ are i.i.d. we have

$$\sum_{i=1}^{m} \ln \frac{1}{V_i} \sim \text{Erlang}(m, \alpha).$$

Therefore:

$$\mathbb{P}\left(\sum_{i=1}^{m} \ln \frac{1}{V_i} \leq \ln\left(\frac{1}{1 - \Delta}\right)\right) = \frac{\alpha^m}{m!} \int_0^{\ln\left(\frac{1}{1-\Delta}\right)} x^{m-1} e^{-\alpha x} dx$$

$$\leq \frac{\alpha^m}{m!} \int_0^{\ln\left(\frac{1}{1-\Delta}\right)} x^{m-1} dx$$

$$= \frac{\alpha^m}{(m)!m} \left[\ln\left(\frac{1}{1 - \Delta}\right)\right]^m.$$

which concludes the proof. $\square$

We state another technical result about the Beta distribution near $1$.

**Lemma 7.** *Consider $V \sim Beta(\alpha + 1, \beta + 1)$ with $\alpha, \beta > 0$. Define $T = \alpha + \beta$ and $M = \frac{\alpha}{\alpha+\beta}$.*

*For all $c \in ]0, 1[$*

$$\mathbb{P}(V \leq c) \leq \frac{e^{\frac{1}{12}}}{\sqrt{2\pi T M(1 - M)}} \int_0^c e^{-TD(M|x)} dx,$$

*with $D$ the Kullback-Leibler divergence between Bernoulli distributions:*

$$D(M \mid x) = M \ln \frac{M}{x} + (1 - M) \ln \frac{1 - M}{1 - x}.$$

*We also have the simpler bound for $c \leq M$ :*

$$\mathbb{P}(V \leq c) \leq \frac{e^{\frac{1}{12}} \sqrt{T}}{\sqrt{2\pi}} e^{-T(M-c)^2}.$$

**Remark 1.** *This proves that for any $\nu \in (0, 1]$*

$$\mathbb{P}\left(V \leq M - \sqrt{\frac{1}{T} \ln\left(\frac{e^{1/12}\sqrt{T}}{\nu\sqrt{2\pi}}\right)}\right) \leq \nu.$$

**Remark 2.** *Consider $V_i \sim Beta(\alpha_i + 1, \beta_i + 1)$ with $V_1, \ldots, V_m$ independent, by negation and union bound we have for all $c \in [0, 1]$ :*

$$\mathbb{P}\left(\sum_{i=1}^{m} V_i \leq \sum_{i=1}^{m} c_i\right) \leq \sum_{i=1}^{m} \mathbb{P}(V_i \leq c_i).$$

*so that the above bound easily extends to the multidimensional case:*

$$\mathbb{P}\left(\sum_{i=1}^{m} V_i \le \sum_{i} M_i - \sqrt{\frac{m^2}{T} \ln\left(\frac{e^{1/12} m \sqrt{T}}{\nu \sqrt{2\pi}}\right)}\right) \le \nu.$$

*Proof.* The density of $V$ is given by:

$$f(x) = \frac{(\alpha+\beta)!}{\alpha!\beta!} x^\alpha (1-x)^\beta.$$

The Stirling approximation yields for all $n$ (see [19])

$$\sqrt{2\pi} n^{n+1/2} \le \sqrt{2\pi} n^{n+1/2} e^{\frac{1}{12n+1}} \le n! \le \sqrt{2\pi} n^{n+1/2} e^{\frac{1}{5n}} \le \sqrt{2\pi} n^{n+1/2} e^{\frac{1}{12}}.$$

Therefore:

$$\frac{(\alpha+\beta)!}{\alpha!\beta!} \le \frac{\sqrt{2\pi}(\alpha+\beta)^{\alpha+\beta+1/2} e^{\frac{1}{12}}}{(2\pi)(\alpha)^{\alpha+1/2}(\beta)^{\beta+1/2}}$$

$$= \frac{T^{T+1/2} e^{\frac{1}{12}}}{\sqrt{2\pi}(TM)^{TM+1/2}(T(1-M))^{T(1-M)+1/2}}$$

$$= \frac{e^{\frac{1}{12}}}{\sqrt{2\pi TM(1-M)}\left[M^M (1-M)^{1-M}\right]^T}.$$

Furthermore:

$$x^\alpha (1-x)^\beta = e^{\alpha \ln(x) + \beta \ln(1-x)}$$

$$= e^{T[M \ln(x) + (1-M) \ln(1-x)]}$$

$$= M^{TM}(1-M)^{(1-M)T} e^{-TD(M|x)}.$$

Replacing:

$$f(x) \le \frac{e^{\frac{1}{12}} e^{-TD(M|x)}}{\sqrt{2\pi TM(1-M)}}.$$

Therefore:

$$\mathbb{P}(V \le c) = \int_0^c f(x) dx \le \frac{e^{\frac{1}{12}}}{\sqrt{2\pi TM(1-M)}} \int_0^c e^{-TD(M|x)} dx.$$

The simpler bound comes from $TM(1-M) = \frac{\alpha\beta}{T} \ge \frac{1}{T}$ and using Pinsker's inequality $D(x \mid M) \ge 2(x-M)^2$, for $c \le M$

$$\mathbb{P}(V \le c) \le \frac{e^{\frac{1}{12}} \sqrt{T}}{\sqrt{2\pi}} e^{-2T(M-c)^2}.$$

$\square$

We recall a result on the Irwin-Hall distribution.

**Remark 3.** *Consider $U_1, ..., U_m$ i.i.d. uniformly distributed in $[0,1]$. Then their sum follows the Irwin-Hall distribution and for any $\Delta \le 1$ we have that:*

$$\mathbb{P}\left(\sum_{i=1}^{m} U_i \ge m - \Delta\right) = \mathbb{P}\left(\sum_{i=1}^{m} U_i \le \Delta\right) = \frac{\Delta^m}{m!}.$$

We present a technical result on the tail behaviour of the sum of beta random variables.

**Lemma 8.** *Consider $V_1, ..., V_m$ independent random variables following beta laws of parameters $(\alpha_1, \beta_1), ..., (\alpha_m, \beta_m)$. For $\epsilon < 1$ we have that :*

$$\mathbb{P}\Big( \sum_{i=1}^{m} V_i \geq m - \epsilon \Big) \leq \frac{\epsilon^{\sum_{i=1}^{m} \beta_i}}{m! \prod_{i=1}^{m} B(\alpha_i, \beta_i)}.$$

*where $B(\alpha, \beta) = \Gamma(\alpha)\Gamma(\beta)/\Gamma(\alpha + \beta)$ is the beta function.*

*Proof.* We define $A_\epsilon \triangleq \{(u_1, ..., u_m) \in [0, 1]^m, m - \epsilon \leq \sum_{i=1}^{m} u_i \leq m\}$. It is noted that if $(u_1, ..., u_m) \in A_\epsilon$ we have that $u_i \geq 1 - \epsilon$ for all $i$. We recall that the probability density of a Beta$(\alpha_i, \beta_i)$ law is $p_i(u) = u^{\alpha_i - 1}(1 - u)^{\beta_i - 1}/B(\alpha_i, \beta_i)$.

We have

$$\mathbb{P}\Big( \sum_{i=1}^{m} V_i \geq m - \epsilon \Big) = \int_{A_\epsilon} \Pi_{i=1}^{m} p_i(u_i) du_1 ... du_m$$

$$= \int_{A_\epsilon} \Pi_{i=1}^{m} \frac{u_i^{\alpha_i - 1}(1 - u_i)^{\beta_i - 1}}{B(\alpha_i, \beta_i)} du_1 ... du_m$$

$$\leq \int_{A_\epsilon} \Pi_{i=1}^{m} \frac{\epsilon^{\beta_i - 1}}{B(\alpha_i, \beta_i)} du_1 ... du_m$$

$$= \frac{\epsilon^{\sum_{i=1}^{m}(\beta_i - 1)}}{\prod_{i=1}^{m} B(\alpha_i, \beta_i)} \int_{A_\epsilon} 1 du_1 ... u_m.$$

But we know that the integral $\int_{A_t} 1 du_1 ... u_m$ corresponds to the cumulative distribution function of the sum of $m$ uniform random variables in $[0, 1]$. This is known as the Irving Hall distribution. So we have that $\int_{A_t} 1 du_1 ... u_m = \frac{\epsilon^m}{m!}$

Which proves the announced result

$$\mathbb{P}\Big( \sum_{i=1}^{m} V_i \geq m - \epsilon \Big) \leq \frac{\epsilon^{\sum_{i=1}^{m} \beta_i}}{m! \prod_{i=1}^{m} B(\alpha_i, \beta_i)}.$$

$\square$

Finally we make an important remark about the link between regret and the first time the optimal decision is selected.

**Remark 4.** *Define $\tau$ the first time the optimal decision is selected. Then we have that:*

$$R(T, \theta) = \mathbb{E}(\sum_{t=1}^{T} \Delta_{x(t)}) \geq \Delta_{\min}\mathbb{E}(\sum_{t=1}^{T} \mathbf{1}\{\Delta_{x(t)} \neq 0\}) \geq \Delta_{\min} \sum_{t=1}^{T} \mathbb{P}(\tau \geq t).$$

## A.2  Proof of Theorem 1

*Proof.* Define $b = 1 - \frac{\Delta}{m}$. Consider $\epsilon > 0$ such that $b - \epsilon \geq \frac{1}{2}$ and denote the two decisions as $x^1 = (1, ..., 1, 0, ..., 0)$ and $x^2 = (0, ..., 0, 1, ..., 1)$. Consider the event where the empirical mean of decision $x^2$ does not deviate too much from its expectation when it is selected:

$$\mathcal{A} = \left\{ \exists t \geq 0 : x(t) = x^2, \ \sum_{i=m+1}^{d} \frac{A_i(t)}{N_i(t)} \leq (b - \epsilon)m \right\}.$$

We decompose $\mathcal{A}$ as $\cup_{n\geq 1}\mathcal{A}_n$ where

$$\mathcal{A}_n = \left\{\exists t \geq 0 : x(t) = x^2, N_i(t) = n, i = m+1, ..., d, \frac{1}{n}\sum_{i=m+1}^{d} A_i(t) \leq (b-\epsilon)m\right\}.$$

Using Hoeffding's inequality we have that:

$$\mathbb{P}(\mathcal{A}) \leq \sum_{n\geq 1}\mathbb{P}(\mathcal{A}_n) \leq \sum_{n\geq 1}\exp(-2mn\epsilon^2) = \frac{\exp(-2m\epsilon^2)}{1-\exp(-2m\epsilon^2)}.$$

where we have used the fact that if $N_i(t) = n$ for $i = m+1, ..., d$ then $\sum_{i=m+1}^{d} A_i(t)$ is a sum of $mn$ i.i.d. Bernoulli variables with parameter $b$. Let us control the probability that decision $x^1$ is never selected between time $0$ and time $t$, which is the probability of event:

$$\mathcal{B}_t = \{x(s) = x^2 : s = 1, ..., t\}.$$

Let us assume that $\mathcal{B}_t$ occurs and $\mathcal{A}$ does not occur. Since decisions $x^1$ and $x^2$ have been selected $0$ and $t$ times respectively, the probability of selecting $x^2$ is lower bounded by:

$$\mathbb{P}(\mathcal{B}_{t+1}|\mathcal{B}_t, \bar{\mathcal{A}}) \geq \mathbb{P}(\sum_{i=1}^{m} V_i(t) \leq \sum_{i=m+1}^{d} V_i(t)|\mathcal{B}_t, \bar{\mathcal{A}}).$$

where $V_1(t), ..., V_d(t)$ are independent, distributed in $[0, 1]$. For $i = 1, ..., m$, $V_i(t)$ is uniformly distributed in $[0, 1]$ and has mean $1/2$. For $i = m+1, ..., d$, $V_i(t)$ has $\mathrm{Beta}(A_i(t)+1, t - A_i(t)+1)$ distribution with mean $\frac{A_i(t)+1}{t+2}$ so that expectations verify:

$$\begin{aligned}
\sum_{i=m+1}^{d}\mathbb{E}(V_i(t)|\mathcal{B}_t, \bar{\mathcal{A}}) - \sum_{i=1}^{m}\mathbb{E}(V_i(t)|\mathcal{B}_t, \bar{\mathcal{A}}) &= \sum_{i=m+1}^{d}\frac{A_i(t)+1}{t+2} - \sum_{i=1}^{m}\frac{1}{2}\\
&\geq \frac{tm(b-\epsilon)+m}{t+2} - \frac{m}{2}\\
&= \frac{mt(b-\epsilon-1/2)}{t+2}\\
&\geq \frac{m(b-\epsilon-1/2)}{3},
\end{aligned}$$

since $\sum_{i=m+1}^{d} A_i(t) \geq tm(b-\epsilon)$.

Using Hoeffding's inequality once again we have:

$$\begin{aligned}
\mathbb{P}\Big(\sum_{i=1}^{m} V_i(t) \geq \sum_{i=m+1}^{d} V_i(t)|\mathcal{B}_t, \bar{\mathcal{A}}\Big) &= \mathbb{P}\Big(\sum_{i=1}^{m} V_i(t) - \sum_{i=m+1}^{d} V_i(t) \geq 0|\mathcal{B}_t, \bar{\mathcal{A}}\Big)\\
&\leq \exp\{-2m(b-\epsilon-1/2)^2/9\}\\
&\equiv p_\Delta.
\end{aligned}$$

We have proven that for all $t > 1$:

$$\mathbb{P}(\mathcal{B}_{t+1}|\mathcal{B}_t, \bar{\mathcal{A}}) \geq 1 - p_\Delta,$$

and since $\mathbb{P}(\mathcal{B}_1|\bar{\mathcal{A}}) = 1/2$:

$$\mathbb{P}(\mathcal{B}_t) \geq \mathbb{P}(\mathcal{B}_t, \bar{\mathcal{A}}) = \mathbb{P}(\bar{\mathcal{A}})\mathbb{P}(\mathcal{B}_t|\bar{\mathcal{A}}) \geq \mathbb{P}(\bar{\mathcal{A}})\mathbb{P}(\mathcal{B}_1|\bar{\mathcal{A}})(1-p_\Delta)^{t-1} = \frac{\mathbb{P}(\bar{\mathcal{A}})}{2}(1-p_\Delta)^{t-1}.$$

Denote by $\tau$ the first time that $x^1$ is selected. If $\mathcal{B}_t$ occurs then $\tau \geq t$ and using Remark 4 yields the lower bound:

$$R(T, \theta) \geq \Delta\sum_{t=1}^{T}\mathbb{P}(\tau \geq t) \geq \frac{\Delta\mathbb{P}(\bar{\mathcal{A}})}{2}\sum_{t=1}^{T}(1-p_\Delta)^{t-1}.$$

Setting $\epsilon = \frac{1}{\sqrt{m}}$ we get that

$$\mathbb{P}(\mathcal{A}) \leq \frac{e^{-2}}{1 - e^{-2}} \leq \frac{1}{2},$$

and we get the announced result:

$$R(T, \theta) \geq \frac{\Delta}{4} \sum_{t=1}^{T} (1 - p_\Delta)^t.$$

$\square$

### A.3 Proof of Theorem 2

*Proof.* Consider $m \geq 5$. We denote by $N^1(t)$ and $N^2(t)$ the number of times that decisions $x^1$ and $x^2$ have been respectively selected, and it is noted that $N_i(t) = N^1(t)$ for $i = 1, ..., m$ and $N_i(t) = N^2(t)$ for $i = m + 1, ..., d$. Consider the event where the empirical mean of decision $x^2$ deviates significantly from its expectation when it is selected:

$$\mathcal{A} = \left\{ \exists t \geq 0 : x(t) = x^2, \frac{1}{N^2(t)} \sum_{i=m+1}^{d} A_i(t) \leq m - \Delta - \sqrt{\frac{m \ln(2N^2(t))}{N^2(t)}} \right\}.$$

We decompose $\mathcal{A}$ as $\cup_{n \geq 1} \mathcal{A}_n$ where

$$\mathcal{A}_n = \left\{ \exists t \geq 0 : x(t) = x^2, N^2(t) = n, \frac{1}{n} \sum_{i=m+1}^{d} A_i(t) \leq m - \Delta - \sqrt{\frac{m \ln(2n)}{n}} \right\}.$$

Using Hoeffding's inequality we have that :

$$\mathbb{P}(\mathcal{A}) \leq \sum_{n \geq 1} \mathbb{P}(\mathcal{A}_n) \leq \sum_{n \geq 1} \frac{1}{(2n)^2} = \frac{\pi^2}{24} \leq \frac{1}{2},$$

where we have used the fact that if $N^2(t) = n$ then $\sum_{i=m+1}^{d} A_i(t)$ is a sum of $mn$ i.i.d. Bernoulli random variables with parameter $1 - \frac{\Delta}{m}$. Let us control the probability that decision $x^1$ is never selected between time $0$ and time $t$, which is the probability of event:

$$\mathcal{B}_t = \{x(s) = x^2 : s = 1, ..., t\}.$$

We have that:

$$\mathbb{P}(\mathcal{B}_{t+1} | \mathcal{B}_t, \bar{\mathcal{A}}) \geq \mathbb{P}\left( \sum_{i=1}^{m} V_i(t) \leq \sum_{i=m+1}^{d} V_i(t) | \mathcal{B}_t, \bar{\mathcal{A}} \right) \geq (1 - p_{t,1})(1 - p_{t,2}),$$

with

$$p_{t,1} = \mathbb{P}\left( \sum_{i=1}^{m} V_i(t) \geq m - \Delta - h(m, t) | \mathcal{B}_t, \bar{\mathcal{A}} \right),$$

$$p_{t,2} = \mathbb{P}\left( \sum_{i=m+1}^{d} V_i(t) \leq m - \Delta - h(m, t) | \mathcal{B}_t, \bar{\mathcal{A}} \right),$$

$$h(m, t) = \sqrt{\frac{m \ln(2t)}{t}} + \sqrt{\frac{m^2}{t} \ln\left( \frac{e^{1/12} m \sqrt{t}}{\frac{1}{t^2} \sqrt{2\pi}} \right)}.$$

It is noted that there exists a universal constant $C_1 > 0$ such that

$$h(m, t) \leq \sqrt{\frac{C_1 m^2 (\ln t + \ln m)}{t}}.$$

Let us define $T_0 = C_0 m^2 \ln m$ with $C_0 > 0$ a universal constant such that the five following inequalities are true:

- $T_0 \geq m$,

- $h(m, t) \leq \frac{1}{3}$ for all $t \geq T_0$,

- $\frac{C_1 8 e^2 \ln t}{t} \leq 1$ for all $t \geq T_0$,

- $\sum_{t=T_0}^{+\infty} \left( \frac{C_1 8 e^2 \ln t}{t} \right)^{\frac{3}{2}} \leq \frac{1}{3}$.

- $\sum_{t=T_0}^{+\infty} \frac{1}{t^2} \leq \frac{1}{2}$.

Consider $p_{t,2}$, and recall that for $i = 1, ..., d$

$$M_i(t) \triangleq \frac{A_i(t)}{A_i(t) + B_i(t)} = \frac{A_i(t)}{N_i(t)}.$$

is the mode of $V_i(t)$. If event $\bar{\mathcal{A}}$ occurs then

$$\sum_{i=m+1}^{d} M_i(t) > m - \Delta - \sqrt{\frac{m \ln(2N^2(t))}{N^2(t)}}.$$

So using lemma 7 and remark 2 we have that:

$$p_{t,2} \leq \frac{1}{t^2}.$$

Consider $p_{t,1}$. Since $\Delta \leq \frac{1}{6}$ and $h(m, t) \leq \frac{1}{3}$ we have

$$m - \Delta - h(m, t) \geq m - \frac{1}{2} \geq m - 1.$$

If event $\mathcal{B}_t$ occurs, then $A_i(t) = B_i(t) = 0$ for all $i = 1, ..., m$ therefore $\sum_{i=1}^{m} V_i(t)$ follows the Irwin-Hall distribution of size $m$, so from remark 3, for $t \geq T_0$ we have:

$$p_{t,1} = \frac{1}{m!} \left( \Delta + h(m, t) \right)^m \leq \frac{1}{m!} \left( (2\Delta)^m + (2h(m, t))^m \right),$$

where we used the convexity inequality $\left( \frac{x+y}{2} \right)^m \leq \frac{x^m + y^m}{2}$.

We have, for $T > T_0$:

$$\frac{\mathbb{P}(\mathcal{B}_T | \bar{\mathcal{A}})}{\mathbb{P}(\mathcal{B}_{T_0} | \bar{\mathcal{A}})} = \prod_{t=T_0}^{T-1} \mathbb{P}(\mathcal{B}_{t+1} | \mathcal{B}_t, \bar{\mathcal{A}}) \geq \prod_{t=T_0}^{T-1} (1 - p_{t,1})(1 - p_{t,2}).$$

Using the union bound and the definition of $T_0$:

$$\prod_{t=T_0}^{T-1} (1 - p_{t,1}) \geq 1 - \sum_{t=T_0}^{T-1} p_{t,1} \geq 1 - \sum_{t=T_0}^{T-1} \frac{1}{t^2} \geq 1 - \sum_{t=T_0}^{+\infty} \frac{1}{t^2} \geq \frac{1}{2}.$$

Now:

$$
\begin{aligned}
1 - p_{t,2} &= 1 - \frac{(2\Delta)^m}{m!} - \frac{(2h(m,t))^m}{m!} \\
&= (1 - \frac{(2\Delta)^m}{m!}) \frac{1 - \frac{(2\Delta)^m}{m!} - \frac{(2h(m,t))^m}{m!}}{1 - \frac{(2\Delta)^m}{m!}} \\
&\geq (1 - \frac{(2\Delta)^m}{m!})(1 - \frac{3}{2} \frac{(2h(m,t))^m}{m!}),
\end{aligned}
$$

where we used the fact that $\Delta \leq \frac{1}{6}$ so that $\frac{(2\Delta)^m}{m!} \leq \frac{1}{3}$.

Using the union bound once more:

$$\prod_{t=T_0}^{T-1}(1-p_{t,2}) \geq \prod_{t=T_0}^{T-1}(1-\frac{(2\Delta)^m}{m!})(1-\frac{3}{2}\frac{(2h(m,t))^m}{m!})$$

$$\geq (1-\frac{(2\Delta)^m}{m!})^{T-T_0}(1-\frac{3}{2}\sum_{t=T_0}^{T-1}\frac{(2h(m,t))^m}{m!})$$

$$\geq (1-\frac{(2\Delta)^m}{m!})^{T-T_0}(1-\frac{3}{2}\sum_{t=T_0}^{\infty}\frac{(2h(m,t))^m}{m!}).$$

We turn to the last sum in the right hand side of the equation above. Since $t \geq T_0 \geq m$ we have

$$h(m,t) \leq \sqrt{\frac{C_1 m^2(\ln t + \ln m)}{t}} \leq \sqrt{\frac{C_1 2m^2 \ln t}{t}}.$$

Using Stirling's approximation we have $m! \geq (m/e)^m$ so that

$$\sum_{t=T_0}^{\infty}\frac{(2h(m,t))^m}{m!} \leq \sum_{t=T_0}^{\infty}\left(\frac{C_1 8e^2 \ln t}{t}\right)^{\frac{m}{2}} \leq \sum_{t=T_0}^{\infty}\left(\frac{C_1 8e^2 \ln t}{t}\right)^{\frac{3}{2}} \leq \frac{1}{3},$$

where we used twice the definition of $T_0$ and $m \geq 5 \geq 3$.

Putting things together we have proven that :

$$\frac{\mathbb{P}(\mathcal{B}_T|\bar{\mathcal{A}})}{\mathbb{P}(\mathcal{B}_{T_0}|\bar{\mathcal{A}})} \geq \frac{1}{4}\left(1-\frac{(2\Delta)^m}{m!}\right)^{T-T_0}.$$

We showed previously with Theorem 1 that :

$$\mathbb{P}(\mathcal{B}_{T_0}|\bar{\mathcal{A}}) \geq \frac{1}{2}(1-p_\Delta)^{T_0-1}$$

Let us lower bound the r.h.s. of this inequality. Since $m \geq 5$ and $\Delta \leq 1/6$ we have, by definition

$$p_\Delta = \exp\left\{-\frac{2m}{9}\left[\frac{1}{2}-\left(\frac{\Delta}{m}+\frac{1}{\sqrt{m}}\right)\right]^2\right\} \leq \exp\left\{-\xi m\right\},$$

with

$$\xi = \frac{2}{9}\left[\frac{1}{2}-\left(\frac{1}{30}+\frac{1}{\sqrt{5}}\right)\right]^2 > 0.$$

Using the definition of $T_0$ this yields

$$\frac{1}{2}(1-p_\Delta)^{T_0-1} \geq \frac{1}{2}(1-e^{-\xi m})^{C_0 m^2 \ln m - 1} \geq \min_{m \geq 5}\left\{\frac{1}{2}(1-e^{-\xi m})^{C_0 m^2 \ln m - 1}\right\} \equiv C_2,$$

where $C_2$ is a universal constant and $C_2 > 0$ since

$$\lim_{m\to\infty}\left\{\frac{1}{2}(1-e^{-\xi m})^{C_0 m^2 \ln m - 1}\right\} = \frac{1}{2} > 0.$$

We have proven that:

$$\mathbb{P}(\mathcal{B}_{T_0}|\bar{\mathcal{A}}) \geq \frac{1}{2}(1-p_\Delta)^{T_0-1} \geq C_2.$$

which gives

$$\mathbb{P}(\mathcal{B}_T) \geq C_2\left(1-\frac{(2\Delta)^m}{m!}\right)^{T-T_0},$$

and applying Remark 4 concludes the proof. □

### A.4  Proof of Corollary 3

*Proof.* Using the same notation as above, we recall that

$$\mathbb{P}(\mathcal{B}_T) \geq C_3 \left(1 - \frac{(2\Delta)^m}{m!}\right)^T.$$

If $\mathcal{B}_T$ occurs, decision $x^1$ is never played, resulting in a regret of $\Delta T$, therefore:

$$R(T,\theta) \geq \Delta T \mathbb{P}(\mathcal{B}_T) \geq C_3 \Delta T \left(1 - \frac{(2\Delta)^m}{m!}\right)^T.$$

(i) If $T \geq 3^m m!$ let us set

$$\Delta = \frac{1}{2}\left(\frac{m!}{T}\right)^{\frac{1}{m}},$$

so that we have $\Delta \leq \frac{1}{6}$ and, using Stirling's approximation $m! \geq (m/e)^m$ we get

$$\max_{\theta \in [0,1]^d} R(T,\theta) \geq \frac{C_3}{3}(m!)^{\frac{1}{m}} T^{1-\frac{1}{m}} \left(1 - \frac{1}{T}\right)^T$$

$$\geq \frac{C_3}{3}\frac{m}{e}(1-e^{-1})T^{1-\frac{1}{m}}.$$

this yields

$$\max_{\theta \in [0,1]^d} R(T,\theta) \geq \mathcal{O}(mT^{1-\frac{1}{m}}).$$

(ii) If $T \leq 3^m m!$ let us set $\Delta = 1/6$, which yields

$$\max_{\theta \in [0,1]^d} R(T,\theta) \geq \mathcal{O}(T).$$

and completes the proof.

$\square$

### A.5  Proof of Theorem 4

*Proof.* To simplify notation, we assume that the $\ell$ rounds of exploration are done before the algorithm starts, so that at time $t = 0$ each decision has been explored $\ell/2$ times and the TS algorithm starts.

We consider the following event :

$$\mathcal{C} = \left\{\forall i \in [d], A_i(0) = \frac{\ell}{2}\right\}.$$

We know that $A_i(0)$, $i = 1, ..., d$ are independent with a Binomial$(\ell/2, \theta_i)$ distribution so that

$$\mathbb{P}(\mathcal{C}) = \left(1 - \frac{\Delta}{m}\right)^{\frac{\ell m}{2}}.$$

Define $\epsilon = \frac{1}{\sqrt{m}}$. We consider again the event where the empirical mean of decision $x^2$ deviates significantly from its expectation when it is selected, accounting for the rounds of forced exploration:

$$\mathcal{A} = \left\{\exists t \geq 0 : x(t) = x^2, \sum_{i=m+1}^{d} A_i(t) \leq (1 - \frac{\Delta}{m} - \epsilon)(N^2(t) - \frac{\ell}{2})m + \frac{\ell}{2}m\right\}.$$

We decompose $\mathcal{A}$ as $\cup_{n \geq 1} \mathcal{A}_n$ where

$$\mathcal{A}_n = \left\{\exists t \geq 0 : x(t) = x^2, N^2(t) = n + \frac{\ell}{2}, \sum_{i=m+1}^{d} A_i(t) \leq (1 - \frac{\Delta}{m} - \epsilon)nm + \frac{\ell}{2}m\right\}$$

Since $\epsilon = \frac{1}{\sqrt{m}}$, using Hoeffding's inequality we have that :

$$\mathbb{P}(\mathcal{A}|\mathcal{C}) \leq \sum_{n \geq 1} \mathbb{P}(\mathcal{A}_n|\mathcal{C}) \leq \sum_{n \geq 1} \exp(-2mn\epsilon^2) = \frac{\exp(-2m\epsilon^2)}{1 - \exp(-2m\epsilon^2)} \leq \frac{1}{2}.$$

where we have used the fact that if $N^2(t) = n + \ell/2$, then $\sum_{i=m+1}^{d} A_i(t)$ equals $\frac{\ell}{2}m$ plus the sum of $mn$ i.i.d Bernoulli random variables with parameter $1 - \frac{\Delta}{m}$. Let us control the probability that decision $x^1$ is never selected between time $0$ and time $t$, which is the probability of event:

$$\mathcal{B}_t = \{x(s) = x^2 : s = 1, ..., t\}.$$

Let us assume that $\mathcal{B}_t$ and $\mathcal{C}$ occurs but $\mathcal{A}$ does not occur. Since decisions $x^1$ and $x^2$ have been selected $\ell/2$ and $\ell/2 + t$ times respectively, the probability of selecting $x^2$ is lower bounded by:

$$\mathbb{P}(\mathcal{B}_{t+1}|\mathcal{B}_t, \bar{\mathcal{A}}, \mathcal{C}) \geq \mathbb{P}\Big(\sum_{i=1}^{m} V_i(t) \leq \sum_{i=m+1}^{d} V_i(t)|\mathcal{B}_t, \bar{\mathcal{A}}, \mathcal{C}\Big).$$

where $V_1(t), ..., V_d(t)$ are independent, distributed in $[0, 1]$. For $i = 1, ..., m$, $V_i(t)$ follows a Beta$(\frac{\ell}{2}, 1)$ law and has mean $\frac{\frac{\ell}{2}+1}{\frac{\ell}{2}+2}$. For $i = m+1, ..., d$, $V_i(t)$ follows a Beta$(A_i(t) + 1, t + \frac{\ell}{2} - A_i(t) + 1)$ distribution with mean $\frac{A_i(t)+1}{t+\frac{\ell}{2}+2}$ so that the expectations verify:

$$\sum_{i=m+1}^{d} \mathbb{E}(V_i(t)|\mathcal{B}_t, \bar{\mathcal{A}}, \mathcal{C}) - \sum_{i=1}^{m} \mathbb{E}(V_i(t)|\mathcal{B}_t, \bar{\mathcal{A}}, \mathcal{C}) \geq m\frac{t(1 - \frac{\Delta}{m} - \epsilon) + \frac{\ell}{2} + 1}{t + \frac{\ell}{2} + 2} - m\frac{\frac{\ell}{2}+1}{\frac{\ell}{2}+2}$$

$$\geq m\frac{(1 - \frac{\Delta}{m} - \epsilon) + \frac{\ell}{2} + 1}{\frac{\ell}{2} + 3} - m\frac{\frac{\ell}{2}+1}{\frac{\ell}{2}+2}$$

$$= m\frac{\left(\frac{1}{\frac{\ell}{2}+2} - (\frac{\Delta}{m} + \epsilon)\right)}{(\frac{\ell}{2} + 3)},$$

since $\sum_{i=m+1}^{d} A_i(t) \geq tm(1 - \frac{\Delta}{m} - \epsilon) + \frac{m\ell}{2}$. Recall that $\epsilon = \frac{1}{\sqrt{m}}$ so that $\frac{1}{\frac{\ell}{2}+2} - (\frac{\Delta}{m} + \epsilon) \geq 0$. Using Hoeffding's inequality:

$$\mathbb{P}\Big(\sum_{i=1}^{m} V_i(t) \geq \sum_{i=m+1}^{d} V_i(t)|\mathcal{B}_t, \bar{\mathcal{A}}, \mathcal{C}\Big) = \mathbb{P}\Big(\sum_{i=1}^{m} V_i(t) - \sum_{i=m+1}^{d} V_i(t) \geq 0\Big)$$

$$\leq \exp\left\{-2m\frac{\left(\frac{1}{\frac{\ell}{2}+2} - (\frac{\Delta}{m} + \epsilon)\right)^2}{(\frac{\ell}{2} + 3)^2}\right\} \equiv p_\Delta^\ell.$$

We have proven that for all $t > 1$:

$$\mathbb{P}(\mathcal{B}_{t+1}|\mathcal{B}_t, \bar{\mathcal{A}}) \geq 1 - p_\Delta^\ell.$$

so that:

$$\mathbb{P}(\mathcal{B}_t) \geq \mathbb{P}(\mathcal{B}_t, \bar{\mathcal{A}}, \mathcal{C})$$
$$= \mathbb{P}(\bar{\mathcal{A}}|\mathcal{C})\mathbb{P}(\mathcal{C})\mathbb{P}(\mathcal{B}_t|\bar{\mathcal{A}}, \mathcal{C})$$
$$\geq \mathbb{P}(\bar{\mathcal{A}}|\mathcal{C})\mathbb{P}(\mathcal{B}_1|\bar{\mathcal{A}}, \mathcal{C})(1 - p_\Delta^\ell)^{t-1}(1 - \frac{\Delta}{m})^{\frac{\ell m}{2}}$$
$$= \frac{\mathbb{P}(\bar{\mathcal{A}}|\mathcal{C})}{2}(1 - p_\Delta^\ell)^{t-1}(1 - \frac{\Delta}{m})^{\frac{\ell m}{2}}.$$

Denote by $\tau$ the first time that $x^1$ is selected. If $\mathcal{B}_t$ occurs then $\tau \geq t$ and using Remark 4 yields the lower bound:

$$R(T, \theta) \geq \Delta \sum_{t=1}^{T} \mathbb{P}(\tau \geq t) \geq \Delta\frac{\mathbb{P}(\bar{\mathcal{A}}|\mathcal{C})}{2}(1 - \frac{\Delta}{m})^{\frac{\ell m}{2}} \sum_{t=1}^{T} (1 - p_\Delta^\ell)^{t-1}.$$

From above, $\mathbb{P}(\mathcal{A}|\mathcal{C}) \leq \frac{1}{2}$, and we get the announced result:

$$R(T, \theta) \geq \frac{\Delta}{4}(1 - \frac{\Delta}{m})^{\frac{\ell m}{2}} \sum_{t=1}^{T} (1 - p_\Delta^\ell)^t.$$

$\square$

### A.6 Proof of Theorem 5

*Proof.* To simplify notation, we assume that the $\ell$ rounds of exploration are done before the algorithm starts, so that at time $t = 0$ each decision has been explored $\ell/2$ times and the TS algorithm starts.

We consider the following event :

$$\mathcal{C} = \left\{ \forall i \in [d], A_i(0) = \frac{\ell}{2} \right\}.$$

We know that $A_i(0)$, $i = 1, ..., d$ are independent with a Binomial$(\ell/2, \theta_i)$ distribution so that

$$\mathbb{P}(\mathcal{C}) = \left(1 - \frac{\Delta}{m}\right)^{\frac{\ell m}{2}}.$$

We consider the event where the empirical mean of decision $x^2$ deviates significantly from its expectation when it is selected, accounting for the rounds of forced exploration:

$$\mathcal{A} = \left\{ \exists t \geq 0 : x(t) = x^2, \sum_{i=m+1}^{d} A_i(t) \leq (m - \Delta)\left(N^2(t) - \frac{\ell}{2}\right) + \frac{\ell m}{2} - \sqrt{m \ln(2(N^2(t) - \frac{\ell}{2}))} \right\}.$$

We decompose $\mathcal{A}$ as $\cup_{n \geq 1} \mathcal{A}_n$ where

$$\mathcal{A}_n = \left\{ \exists t \geq 0 : x(t) = x^2, N^2(t) = n + \frac{\ell}{2}, \sum_{i=m+1}^{d} A_i(t) \leq (m - \Delta)n + \frac{\ell m}{2} - \sqrt{m \ln(2n)} \right\}.$$

Using Hoeffding's inequality we have that :

$$\mathbb{P}(\mathcal{A}|\mathcal{C}_\ell) \leq \sum_{n \geq 1} \mathbb{P}(\mathcal{A}_n|\mathcal{C}_\ell) \leq \sum_{n \geq 1} \frac{1}{(2n)^2} = \frac{\pi^2}{24} \leq \frac{1}{2},$$

where we have used the fact that if $N^2(t) = n + \ell/2$, then $\sum_{i=m+1}^{d} A_i(t)$ equals $\frac{\ell}{2}m$ plus the sum of $mn$ i.i.d Bernoulli random variables with parameter $1 - \frac{\Delta}{m}$. Let us control the probability that decision $x^1$ is never selected between time $0$ and time $t$, which is the probability of event:

$$\mathcal{B}_t = \{x(s) = x^2 : s = 1, ..., t\},$$

We have that:

$$\mathbb{P}(\mathcal{B}_{t+1}|\mathcal{B}_t, \bar{\mathcal{A}}) \geq \mathbb{P}\left( \sum_{i=1}^{m} V_i(t) \leq \sum_{i=m+1}^{d} V_i(t)|\mathcal{B}_t, \bar{\mathcal{A}} \right) \geq (1 - p_{t,1})(1 - p_{t,2}),$$

with

$$p_{t,1} = \mathbb{P}\left( \sum_{i=1}^{m} V_i(t) \geq m - \Delta - h(m, \ell, t)|\mathcal{B}_t, \bar{\mathcal{A}}, \mathcal{C}_\ell \right),$$

$$p_{t,2} = \mathbb{P}\left( \sum_{i=m+1}^{d} V_i(t) \leq m - \Delta - h(m, \ell, t)|\mathcal{B}_t, \bar{\mathcal{A}}, \mathcal{C}_\ell \right),$$

$$h(m, \ell, t) = -\frac{m\ell}{2t} + \sqrt{\frac{m \ln(2t)}{t}} + \sqrt{\frac{m^2}{t + \ell} \ln\left( \frac{e^{1/12} m \sqrt{t + \ell}}{\frac{1}{t^2} \sqrt{2\pi}} \right)}.$$

It is noted that there exists a constant $C_1 \geq 0$ such that

$$h(m, \ell, t) \leq \sqrt{\frac{C_1 m^2 (\ln m + \ln(t + \ell))}{t}}.$$

Let us define $T_0(m, \ell) = C_0 m^2 (\ln m) \ell^{\frac{1}{4} - \frac{1}{m}}$ with $C_0$ a universal constant such that the following inequalities are true

- $T_0 \geq \max(m, \ell, 7)$,

- $h(m, \ell, t) \leq \frac{1}{6\ell}$, for all $t \geq T_0$,

- $\sum_{t=T_0}^{+\infty} \frac{1}{t^2} \leq \frac{1}{2}$,

- $4\left(\frac{\ell e \sqrt{2C_1}}{(T_0-1)^{\frac{1}{4}-\frac{1}{m}}}\right) \leq \frac{1}{3}$.

First consider $p_{t,2}$, and recall that $\forall i \in [d], \forall t, M_i(t) \triangleq \frac{A_i(t)}{A_i(t)+B_i(t)} = \frac{A_i(t)}{N_i(t)}$ is the mode of $V_i(t)$. If event $\bar{\mathcal{A}}$ occurs then

$$\sum_{i=m+1}^{d} M_i(t) > m - \Delta - \sqrt{\frac{m \ln(2N^2(t))}{N^2(t)}} + \frac{m\ell}{2N^2(t)}.$$

So using lemma 7 and remark 2 we have that: $p_{t,2} \leq \frac{1}{t^2}$.

Consider $p_{t,1}$. Since $\Delta < \frac{1}{6\ell}$ and for $t \geq T_0$ we have $h(m, \ell, t) \leq \frac{1}{6\ell}$, hence

$$m - \Delta - h(m, \ell, t) \geq m - \frac{1}{3\ell}.$$

If event $\mathcal{B}_t$ and $\mathcal{C}_t$ occurs then $A_i(t) = \frac{\ell}{2}$ and $B_i(t) = 0$ for all $i = 1, ..., m$ therefore we may control the tail behaviour of $\sum_{i=m+1}^{d} V_i(t)$ thanks to lemma 8. So we have for $t \geq T_0$:

$$p_{t,1} \leq \frac{\ell^m}{2^m m!} (\Delta + h(m, \ell, t))^m \leq \frac{1}{2(m!)} \left((\ell\Delta)^m + (\ell h(m, \ell, t))^m\right).$$

where we used the convexity inequality $\left(\frac{x+y}{2}\right)^m \leq \frac{x^m+y^m}{2}$.

We have, for $T > T_0$:

$$\frac{\mathbb{P}(\mathcal{B}_T | \bar{\mathcal{A}}, \mathcal{C})}{\mathbb{P}(\mathcal{B}_{T_0} | \bar{\mathcal{A}}, \mathcal{C})} = \prod_{t=T_0}^{T-1} \mathbb{P}(\mathcal{B}_{t+1} | \mathcal{B}_t, \bar{\mathcal{A}}, \mathcal{C}) \geq \prod_{t=T_0}^{T-1} (1 - p_{t,1})(1 - p_{t,2}).$$

Using the union bound and the definition of $T_0$:

$$\prod_{t=T_0}^{T-1} (1 - p_{t,2}) \geq 1 - \sum_{t=T_0}^{T-1} p_{t,2} \geq 1 - \sum_{t=T_0}^{T-1} \frac{1}{t^2} \geq 1 - \sum_{t=T_0}^{+\infty} \frac{1}{t^2} \geq \frac{1}{2}.$$

Now:

$$
\begin{aligned}
1 - p_{t,1} &= 1 - \frac{(\ell\Delta)^m}{m!} - \frac{(\ell h(m, \ell, t))^m}{m!} \\
&= \left(1 - \frac{(\ell\Delta)^m}{m!}\right) \frac{1 - \frac{(\ell\Delta)^m}{m!} - \frac{(\ell h(m,\ell,t))^m}{m!}}{1 - \frac{(\ell\Delta)^m}{m!}} \\
&\geq \left(1 - \frac{(\ell\Delta)^m}{m!}\right)\left(1 - \frac{3}{2}\frac{(\ell h(m, \ell, t))^m}{m!}\right).
\end{aligned}
$$

where we used the fact that $\Delta \leq \frac{1}{6}$ so that $\frac{(\ell\Delta)^m}{m!} \leq \frac{1}{3}$.

Using the union bound once more:

$$
\begin{aligned}
\prod_{t=T_0}^{T-1} (1 - p_{t,2}) &\geq \prod_{t=T_0}^{T-1} \left(1 - \frac{(\ell\Delta)^m}{m!}\right)\left(1 - \frac{3}{2}\frac{(\ell h(m, \ell, t))^m}{m!}\right) \\
&\geq \left(1 - \frac{(\ell\Delta)^m}{m!}\right)^{T-T_0}\left(1 - \frac{3}{2}\sum_{t=T_0}^{T-1}\frac{(\ell h(m, \ell, t))^m}{m!}\right) \\
&\geq \left(1 - \frac{(\ell\Delta)^m}{m!}\right)^{T-T_0}\left(1 - \frac{3}{2}\sum_{t=T_0}^{\infty}\frac{(\ell h(m, \ell, t))^m}{m!}\right).
\end{aligned}
$$

We turn to the last sum in the right hand side of the equation above. It is noted that $\log(2t) \le \sqrt{t}$ for all $t \ge 7$. Since $t \ge T_0 \ge \max(m, \ell, 7)$ we have

$$h(m, \ell, t) \le \sqrt{\frac{C_1 m^2 (\ln(t + \ell) + \ln m)}{t}} \le \sqrt{\frac{C_1 2m^2 \ln(2t)}{t}} \le m\sqrt{2C_1} \; t^{-\frac{1}{4}}.$$

We now upper bound the sum as follows:

$$\sum_{t=T_0}^{\infty} \frac{(\ell h(m, t))^m}{m!} \overset{(i)}{\le} \frac{(\ell m \sqrt{2C_1})^m}{m!} \sum_{t=T_0}^{\infty} t^{-\frac{m}{4}} \overset{(ii)}{\le} (\ell e \sqrt{2C_1})^m \sum_{t=T_0}^{\infty} t^{-\frac{m}{4}}$$

$$\overset{(iii)}{\le} 4 \left( \frac{\ell e \sqrt{2C_1}}{(T_0 - 1)^{\frac{1}{4} - \frac{1}{m}}} \right)^m \overset{(iv)}{\le} 4 \left( \frac{\ell e \sqrt{2C_1}}{(T_0 - 1)^{\frac{1}{4} - \frac{1}{m}}} \right) \le \frac{1}{3},$$

where we used (i) the bound above, (ii) Stirling's approximation $m! \ge (m/e)^m$, (iii) the following sum-integral comparison, for any $m \ge 5$:

$$\sum_{t=T_0}^{+\infty} t^{-\frac{m}{4}} \le \int_{T_0 - 1}^{+\infty} t^{-\frac{m}{4}} \, dt = \frac{(T_0 - 1)^{1 - \frac{m}{4}}}{\frac{m}{4} - 1} \le 4(T_0 - 1)^{1 - \frac{m}{4}},$$

and (iv) the definition of $T_0$.

Putting things together we have proven that

$$\frac{\mathbb{P}(\mathcal{B}_T | \bar{\mathcal{A}}, \mathcal{C})}{\mathbb{P}(\mathcal{B}_{T_0} | \bar{\mathcal{A}})} \ge \frac{1}{4} \left( 1 - \frac{(\ell \Delta)^m}{m!} \right)^{T - T_0}.$$

We showed previously that

$$\mathbb{P}(\mathcal{B}_{T_0} | \bar{\mathcal{A}}) \ge \frac{1}{2} (1 - p_\Delta)^{T_0 - 1} \ge C_2(\ell, m),$$

with

$$C_2(\ell, m) = \frac{1}{2} (1 - p_\Delta^\ell)^{T_0 - 1},$$

and it is noted that $\lim_{m \to \infty} C_2(\ell, m) = 1$ for any fixed $\ell \ge 0$.

Therefore

$$\mathbb{P}(\mathcal{B}_T | \bar{\mathcal{A}}, \mathcal{C}) \ge \frac{C_2(\ell, m)}{4} \left( 1 - \frac{(\ell \Delta)^m}{m!} \right)^{T - T_0}.$$

Since $\mathbb{P}(\mathcal{C}) = (1 - \frac{\Delta}{m})^{\frac{\ell m}{2}}$ we get

$$\mathbb{P}(\mathcal{B}_T | \bar{\mathcal{A}}) \ge \frac{C_2(\ell, m)}{4} (1 - \frac{\Delta}{m})^{\frac{\ell m}{2}} \left( 1 - \frac{(\ell \Delta)^m}{m!} \right)^{T - T_0},$$

and applying Remark 4 concludes the first part of the proof.

Now choose:

$$\Delta = \frac{1}{\ell} \min \left( \left( \frac{m!}{T} \right)^{\frac{1}{m}}, \frac{1}{6} \right),$$

and lower bound the regret by

$$\max_{\theta \in [0,1]^d} R(T, \theta) \ge \Delta T \mathbb{P}(\mathcal{B}_T).$$

If $\Delta = \frac{1}{\ell} \left( \frac{m!}{T} \right)^{\frac{1}{m}}$, then replacing

$$\max_{\theta \in [0,1]^d} R(T, \theta) \ge \Delta T \frac{C_2(\ell, m)}{4} \left( 1 - \frac{\Delta}{m} \right)^{\frac{\ell m}{2}} \left( 1 - \frac{(\ell \Delta)^m}{m!} \right)^T$$

$$= \frac{(m!)^{\frac{1}{m}} T^{1 - \frac{1}{m}}}{\ell} \frac{C_2(\ell, m)}{4} \left( 1 - \frac{\Delta}{m} \right)^{\frac{\ell m}{2}} \left( 1 - \frac{1}{T} \right)^T.$$

Using the facts that (i) $\left(1 - \frac{1}{T}\right)^T \geq e$, that (ii) $(m!)^{\frac{1}{m}} \geq \frac{m}{e}$ which follows from Stirling's approximation $m! \geq \left(\frac{m}{e}\right)^m$ and that (iii) since $\Delta \leq \frac{1}{6\ell}$:

$$\left(1 - \frac{\Delta}{m}\right)^{\frac{\ell m}{2}} \geq \left(1 - \frac{1}{6m\ell}\right)^{\frac{\ell m}{2}} \geq e^{-\frac{1}{12}},$$

which yields the minimax regret bound

$$\max_{\theta \in [0,1]^d} R(T, \theta) \geq \mathcal{O}\left(C_2(\ell, m)\frac{m}{\ell}T^{1-\frac{1}{m}}\right).$$

Otherwise $\Delta = \frac{1}{6\ell}$ and we simply have

$$\max_{\theta \in [0,1]^d} R(T, \theta) \geq \mathcal{O}\left(C_2(\ell, m)\frac{T}{\ell}\right).$$

which completes the proof. □

## B  Linear Bandits: Regret Upper Bound for ESCB

We first recall a regret upper bound for ESCB found in [8], based on the more general analysis of [9].

**Theorem 9.** *Consider a linear combinatorial bandit problem.*

*Then the regret of ESCB is upper bounded by:*

$$R(T, \theta) \leq C(m) + \frac{2dm^3}{\Delta_{\min}^2} + \frac{24d(\ln T + 4m \ln \ln T)}{\Delta_{\min}} \left\lceil \frac{\ln m}{1.61} \right\rceil^2,$$

*with $C(m)$ a positive number that depends solely on $m$.*

## C  Non linear Combinatorial bandits

### C.1  Non-Linear Combinatorial Bandits without Forced Exploration

We here provide a non-linear combinatorial bandits example. The example is inspired by [21]: there are two decisions, the optimal decision has an expected reward of 1 and the other one an expected reward of $1 - \Delta$. Theorem 10 shows that the regret of TS for this problem scales super-exponentially with the dimension $d$ which is an improvement over [21][Theorem 3]. By corollary, we prove that TS does not outperform random choice (i.e. a trivial algorithm which chooses one of the two decisions uniformly at random at each time) until $t \geq T_0(m)$, where $T_0(m)$ grows super-exponentially with $m$, As an illustration of how large this number might be, for $\Delta = \frac{1}{2}$, the value of $T_0(9)$ is greater than a million, and the value of $T_0(20)$ is greater than the estimated age of the universe in seconds. Therefore, in practice as well as in theory, TS does not outperform random choice in high dimensions which is perhaps even more surprising.

The proof of Theorem 10 is based on the fact that there exists a non zero probability that the optimal decision will never be selected for an exponentially large amount of time. Indeed, if the optimal decision has never been selected, it is chosen with a probability equal to $\mathbb{P}(\prod_{i=1}^m U_i \geq 1 - \Delta)$ where $U_1, ..., U_m$ are i.i.d. uniformly distributed on $[0, 1]$, and since this probability is exponentially small in $d$, one must wait for an exponentially large time before selecting the optimal decision and the regret must scale accordingly. It is noted that this proof technique of lower bounding the expected value of the first time the optimal decision is ever selected is very powerful and will be used many times to prove our results.

**Theorem 10.** *Consider a non-linear combinatorial bandit problem over combinatorial set $\mathcal{X} = \{\sum_{i=1}^m e_i, e_{m+1}\}$ where $(e_i)_{i \in [m+1]}$ is the canonical base of $\mathbb{R}^{m+1}$ with parameter $\theta = (1, ..., 1)$ and reward function $f(x, \theta) = \prod_{i=1}^m \theta_i$ if $x = \sum_{i=1}^m e_i$ and $f(x, \theta) = 1 - \Delta$ otherwise.*

*Then the regret of TS is lower bounded by*

$$R(T, \theta) \geq \frac{\Delta}{p_\Delta}(1 - (1 - p_\Delta)^T) \text{ with } p_\Delta = \frac{1}{mm!}\left[\ln\left(\frac{1}{1-\Delta}\right)\right]^m.$$

**Corollary 11.** *For any $T \leq T_0(m) \equiv \frac{1}{p_\Delta}$ TS performs strictly worse than random choice in the sense that*

$$R(T, \theta) \geq T\Delta \left(1 - \frac{1}{e}\right) > \frac{T\Delta}{2}.$$

It is noted that Theorem 10 is a parameter-dependent lower bound, where we consider a fixed parameter $\theta$ and we let the time horizon $T$ grow. From Theorem 10 we deduce Corollary 12 which is a lower bound on the minimax regret of TS. The minimax regret of TS scales at least as $\Omega(T^{1-\frac{1}{d}})$, so that it is almost linear in high dimensions when $d$ is large. This also proves that, as long as the dimension $d$ is strictly greater than 2, TS is not minimax optimal, since there exists algorithms such as CUCB whose minimax regret scales at most as $O(\mathbf{poly}(d)\sqrt{T \ln T})$. This demonstrates that TS has a tendency to be too "greedy" which prevents it from exploring enough, and while this is not a problem in low dimensions, in high dimensions this matters a great deal, and causes it to perform much worse than optimistic algorithms. Corollary 12 is proven simply by letting $\Delta = T^{-\frac{1}{d}}$ in Theorem 10 and the regret upper bound for CUCB follows directly from [14].

**Corollary 12.** *Consider $\mathcal{F}$ the class of $1$-Lipschitz functions.*

*The minimax regret of TS is lower bounded by:*

$$\max_{\theta \in [0,1]^d, f \in \mathcal{F}} R(T, \theta, f) \geq C_1 T^{1-\frac{1}{d}},$$

*with $C_1 > 0$ a universal constant, the minimax regret of CUCB is upper bounded by*

$$\max_{\theta \in [0,1]^d, f \in \mathcal{F}} R(T, \theta, f) \leq C_1' d\sqrt{T \ln T},$$

*where $C_1'$ is a universal constant. Hence TS is* not *minimax optimal.*

## C.2 Proof of Theorem 10

*Proof.* At round $t \geq 1$, if the optimal decision has never been played then $A_i(t) = B_i(t) = 0$ for $i = 1, ..., m$. In turn the samples $V_i(t)$ are independent and uniformly distributed in $[0, 1]$ for $i = 1, ..., m$.

Lemma 6 shows that at time $t$ the optimal decision is played with probability

$$\mathbb{P}\left(\prod_{i=1}^{m} V_i(t) \geq 1 - \Delta\right) \leq p_\Delta \equiv \frac{1}{mm!}\left[\ln\left(\frac{1}{1-\Delta}\right)\right]^m.$$

So the distribution of the first time the optimal decision is played

$$\tau = \min\{t \geq 1 : x(t) = x^\star\},$$

is lower bounded by a geometric law

$$\mathbb{P}(\tau \geq t) \geq (1 - p_\Delta)^{t-1}.$$

Combining this with Remark 4 yields the announced regret bound

$$R(T, \theta) \geq \Delta \sum_{t=1}^{T} \mathbb{P}(\tau \geq t) \geq \sum_{t=1}^{T} (1 - p_\Delta)^{t-1} = \frac{\Delta}{p_\Delta}(1 - (1 - p_\Delta)^T).$$

$\square$

## C.3 Non-Linear Combinatorial Bandits with Forced Exploration

Our results above show that the regret of TS scales exponentially in the dimension since the expectation of the first time at which the optimal decision is selected can grow exponentially in the dimension. Therefore it is natural to assume that forcing some exploration initially would alleviate the problem. Theorem 13 considers the same non linear bandit problem as that considered in Theorem 10, and shows that, while forced exploration does bring some improvement, for any fixed value of $\ell > 0$, the

regret of TS with $\ell$ forced exploration rounds still scales exponentially in the dimension. The reason for this is that once again the first time at which the optimal decision is selected can be exponentially large, even with forced exploration. Upon closer inspection of Theorem 13, one can see that, in order for the regret lower bound not to scale exponentially in the dimension one would require $1/p_\Delta^\ell$ to grow at most polynomially in $d$, which in turn would require $\frac{\ell}{2} \ln\left(\frac{1}{1-\Delta}\right) \geq 1$. This indicates that, unless the gap $\Delta$ is known in advance (and in general $\Delta$ is of course unknown), it is not possible to select a value of $\ell$ that prevents the regret from scaling exponentially in the dimension. This suggests that some more complex modifications need to be made to TS in order to "fix" this exponential dependency on the dimension.

**Theorem 13.** *Consider a non-linear combinatorial bandit problem with over combinatorial set $\mathcal{X} = \{\sum_{i=1}^m e_i, e_{m+1}\}$ with parameter $\theta = (1, ..., 1)$ and reward function $f(x, \theta) = \prod_{i=1}^m \theta_i$ if $x = \sum_{i=1}^m e_i$ and $f(x, \theta) = 1 - \Delta$ otherwise.*

*Then the regret of TS with $\ell$ forced exploration rounds is lower bounded by*

$$R(T, \theta) \geq \frac{\Delta}{p_\Delta^\ell}(1 - (1 - p_\Delta^\ell)^T) \text{ with } p_\Delta^\ell = \frac{1}{mm!}\left[\left(1 + \frac{\ell}{2}\right) \ln\left(\frac{1}{1-\Delta}\right)\right]^m.$$

### C.4 Proof of Theorem 13

*Proof.* At round $t \geq 1$, if the optimal decision has been played $\frac{\ell}{2}$ times then $A_i(t) = \frac{\ell}{2}$ and $B_i(t) = 1$ for $i = 1, ..., m$. In turn the samples $V_i(t)$ are independent with distribution $\text{Beta}(1 + \frac{\ell}{2}, 1)$ for $i = 1, ..., m$.

Lemma 6 shows that at time $t$ the optimal decision is played with probability

$$\mathbb{P}(\prod_{i=1}^m V_i(t) \geq 1 - \Delta) \leq p_\Delta^\ell \equiv \frac{1}{mm!}\left[\left(1 + \frac{\ell}{2}\right) \ln\left(\frac{1}{1-\Delta}\right)\right]^m.$$

So the distribution of the first time the optimal decision is played

$$\tau = \min\{t \geq 1 : x(t) = x^\star\},$$

is lower bounded by a geometric law

$$\mathbb{P}(\tau \geq t) \geq (1 - p_\Delta^\ell)^{t-1}.$$

Combining this with remark 4 yields the announced regret bound

$$R(T, \theta) \geq \Delta \sum_{t=1}^T \mathbb{P}(\tau \geq t) \geq \sum_{t=1}^T (1 - p_\Delta^\ell)^{t-1} = \frac{\Delta}{p_\Delta^\ell}(1 - (1 - p_\Delta^\ell)^T).$$

$\square$