# OpenReview forum: "On the Suboptimality  of Thompson Sampling in High Dimensions"
_NeurIPS.cc/2021/Conference — NeurIPS 2021 Poster_

### Official Review · Reviewer_1SPf · 2021-06-28

**Rating:** 7
**Confidence:** 4

**Summary:**

This paper shows that the Thompson Sampling (TS) is suboptimal for combinatorial semi-bandits. In particular, they give instances where the regret scales exponentially in the ambient dimension and shows that its minimax regret is nearly linear. They demonstrate this phenomenon theoretically and empirically.

**Limitations And Societal Impact:**

The authors do not comment on the potential negative societal impact of their work. I think the paper is mainly theoretical. Algorithms for combinatorial semibandits could be used for applications with either positive societal impact or negative societal impact.

**Main Review:**

I think that the the theoretical and empirical results on the suboptimality of TS are original and surprising. This is significant, as the authors point out, because TS has been analyzed in several recent papers for combinatorial semi-bandits and this suggests that new algorithmic ideas are needed.

The authors provide a nice empirical study on the suboptimality of TS.

In the sketch of the proof following Theorem 1, the authors provide intuition on the large t regime at which point the law of large numbers kicks in. It would be useful to explain in addition why TS does not sample the best decision with sufficiently large probability in the small t regime.  This seems to be a gap currently in the proof sketch.

I think it would be useful if they provided a regret bound for ESCB for Theorem 1 to emphasize that there exists an algorithm whose regret does not scale exponentially in the ambient dimension.

In section 3.4, the authors write that the learner would need to know the value of the gap in advance to set l. But, could one use the doubling trick, as well as perhaps a confidence bound?

One limitation of the current paper is that they do not give any positive results. It would be really nice if they could modify TS in a way that avoids this poor scaling in the dimension, even if this were just a partial result. I think that this would make the paper considerably stronger.

After rebuttal: I found the authors' response convincing, and have increased my score.


**Time Spent Reviewing:**

3

---

> ### Author Response · Authors · 2021-08-09
> **Reviewer response**
>
> Dear reviewer,
>
> - Regret bound for optimistic algorithms: we propose to state the best currently known regret bounds for optimistic algorithms (for instance ESCB) to allow for an easy comparison with Thompson sampling.
> - Proof sketch of Theorem one : In the small $T$ regime, the probability of sampling the optimal decision is approximately $1/2$. It is the probability of sampling the optimal decision at time $t=1$ because the priors are uniform. In fact our analysis shows that there are two kinds of sample paths: either Thompson plays the optimal decision in the first rounds and then everything works out well, otherwise it never plays it, resulting in an overly greedy behavior and a very large regret on those sample paths.
> - Doubling Trick : Indeed, reintroducing forced explorations from time to time (at each epoch of a doubling trick for instance), seems like an interesting idea for forcing exploration. However it is not completely obvious whether or not this alleviates the problem and is a possible idea for future work.

---

### Official Review · Reviewer_dwfb · 2021-07-08

**Rating:** 7
**Confidence:** 3

**Summary:**

This paper focusses on Thompson Sampling for stochastic combinatorial semibandits. The main contributions of the paper are to formalise conditions under which Thompson Sampling does not behave optimally for these problems. Through per-instance and minimax analyses, the authors show that when Thompson Sampling is deployed for Bernoulli combinatorial semibandits, and is initialised with a uniform (Beta(1,1)) prior there exist particular action sets and underlying parameters of the reward distribution such that the regret grows sub-optimally. In particular the regret can be shown to be lower bounded by a function that is exponential in the number of arms, or linear in the time horizon for small enough (but potentially still large) time horizons. An empirical investigation ratifies these findings and demonstrates that this issue is not encountered by optimistic algorithms such as ESCB and CUCB.

**Limitations And Societal Impact:**

Yes, the authors have considered this.

**Main Review:**

I feel there is a solid piece of theoretical analysis underpinning this work, and that the experimental work has been thoughtfully conducted, resulting in a paper which does a good job of investigating an interesting phenomenon. I believe this a useful finding, of genuine interest to the NeurIPS community. However, I do also feel that there are some issues which should be addressed to improve the paper. These are discussed in more detail below, but principally concern the dependence of the results on a particular choice of prior and construction of the action set, and the presentation of the proofs of Theorem 2 and Corollary 3.

ORIGINALITY: The paper presents new and interesting theoretical results. The key result underpinning these is a lower bound (in terms of a horizon T) on the probability of not selecting the optimal action within T rounds, when playing the particular combinatorial problem with action set X^p introduced in section 3. This result is as far as I am aware new, and correct. I feel that it tells us something concrete about Thompson Sampling that we, as a community, only had intuition- or simulation-based evidence for previously, and thus I am happy that this paper clears the bar for originality.

QUALITY: The writing and the research is generally of a high quality. The introductory text displays a good combination of concise mathematics and a clear statement of the main ideas of the paper. The quality of the writing drops somewhat in the proofs – at points reaching a minimal level of detail. In particular in the proofs of Theorem 2 and Corollary 3 (and the results in a similar style to these) there are a couple of key points that are not as clear as they could be due to this succinct writing style. I feel the proof would be improved if the authors could add detail to clarify the following points: Q1: In line 535, in what sense is the constant C_2 universal? In that it does not depend on \Delta? Or  T? or m? It seems that a dependence on m at least is inevitable? Seeing as the final bounds include dependence on m should this not be stated? Q2: In line 536, how does the bound on P(B_T) follow from a bound on P(B_T|\bar{A})? This is not discussed.  Q3: Similarly to Q1, on line 539, is C_3 also a universal constant, and what is its dependence on m, via both T_0 and C_2? Further, statements such as that on line 534 would be much more user friendly if they referenced a particular equation rather than using a phrase such as “We showed previously”.

I wish to praise the quality of the experimental section. I feel that the authors have chosen sensible experiments to usefully interface with the theory presented.

CLARITY: As above, the clarity of the writing is generally clear, dropping somewhat in the appendices – I would react positively to a general attempt to include more detail throughout the proofs to aid the less senior reader. An issue I do have, that best falls under the heading of clarity is the context in which the results have been presented. I feel that the generality of the results is tending towards being oversold. The tone of the first few sections suggests that a general issue with Thompson Sampling as a principle which may apply widely across combinatorial bandits has been identified. I feel this is stretching the contribution somewhat and downplaying the nature of the results which to me represent the identification of pathological cases. I wish to make clear that I have no issue with the presentation of such results, and no issue with the acceptance of such results to NeurIPS. However, I think it is important to make clear(er) from the outset that these results apply to a) Thompson Sampling with the uniform prior, and b) are only verified for a particular formulation of the action set – which the informed reader may well observe could be more parsimoniously represented as a 2-armed bandit with rewards in the discrete set {0,1,…,m}, where, from what I can see, a traditional instantiation of Thompson Sampling would not inherit this suboptimal performance? I would like to see a commitment to a more transparent framing of these issues – particular the issue around the choice of prior, which I think is an important consideration. I am aware of at least one study (Grant et al, 2020 – Adaptive Policies for Perimeter Surveillance) which investigates empirically the effect of the prior on Thompson Sampling in an instance of a combinatorial bandit, and a number considering the effect of the prior in more classic multi-armed bandit settings (Bubeck and Liu, 2014 – Prior-free and Prior-dependent regret bounds for Thompson Sampling, Liu and Li, 2015 - On the Prior Sensitivity of Thompson Sampling) which may be useful references in speculating as to the effect of the prior. My particular suspicion is that a more 'optimistic' prior may alleviate the suboptimal performance.

SIGNFICANCE: I think this is paper provides a set of useful observations around the performance of Thompson Sampling on combinatorial bandits, and modulo some achievable improvements to the presentation of the results certainly significant and interesting to the NeurIPS community. The authors may wish to add a more extensive history in terms of the observation of the undesirable phenomenon, noting that (Agrawal et al, 2017 – ‘Thompson Sampling for the MNL-bandit’) already propose a fix to the issue through using correlated Thompson samples to increase the probability of exploration in a very closely related problem.


MISC: Finally, I would suggest that a thorough copy-edit be conducted prior to upload of any final version as I found a number of grammatical errors (e.g. line 110 – should be ‘performs’, line 230 should be ‘occasions’, line 267 should be ‘concerns’, Fig 1 pt 5 – should be ‘receives’, line 541 is not a sentence incorporated properly in to the text), and typos (e.g. line 134 – referencing issue, Figs 6 and 7 – should be X^p in part (a)s, line 535 – spacing between equations needed), and places where resizing of nested parentheses would aid the reading (e.g. lines 147, 185, and 204).


**Time Spent Reviewing:**

6

---

> ### Author Response · Authors · 2021-08-09
> **Reviewer response**
>
> Dear reviewer,
>
> Throughout the main paper and the proof, we call a "universal constant" a number that does not depend on $T$, $m$, $d$, $\Delta$, etc.  We very much agree with the remarks in the "Clarity" section, for instance the fact that our work involved identifying pathological cases as well as the impact of priors on performance, and we will modify the presentation of the results to reflect this. We will make the proof more clear and self contained by filling the gaps highlighted by the reviewer and clearly referencing previous results.
>
> On the topic of optimistic priors and the link with forced exploration: We do agree that setting a more 'optimistic prior' can help with performance. However if one looks at the forced exploration case and in particular the event : $C_l$ line 572.  One can suppose that event $C_l$ happens which is the case at the start of our analysis. It is the same as setting the prior distribution for each arm to $\text{Beta}(l/2+1,1)$ that has a mean of ${l/2 + 1 \over l/2+2}$ which is very near to 1.  Setting a prior like this one does not alleviate the problem of linear minimax regret. (We do agree however that this happen for very small gap $\Delta$). Therefore we believe that prior selection could be non trivial.
>
> We address your questions below:
>
> Q1 : $C_2$ line 535 is indeed a universal constant, so that it does not depend on $T,m,d,\Delta$. While this may seem counter intuitive, this is true by the following reasoning.
>
> There is a minor typo in the statement of Theorem 2 and its proof: the assumption $m \ge 3$ should be replaced by $m \ge 5$. Everything else holds verbatim.
>
> Recall that:
> - $p_{\Delta} = \exp[-{2m \over 9} ( {1 \over 2} - ( {\Delta \over m} + {1 \over \sqrt{m}}))^2 ]$
> - $T_0 = C_0 m^2\ln m$ with $C_0$ a universal constant
> - we assume that $m \ge 5$
> - we consider the small gap regime where $\Delta< 1/6$.
>
> Since $m \ge 5$ and $\Delta \ge 1/6$ we have:
>
> $ {\Delta \over m} + {1 \over \sqrt{m}}  \le {1 \over 30} + {1 \over \sqrt{5}}$
>
> Replacing in the definition of $p_{\Delta}$ we have
>
> $
> p_{\Delta} \le e^{- a m}
> $
>
> where $a > 0$ is a universal constant defined as
>
> $
> a = { 2 \over 9 } [ {1 \over 2} - (  {1 \over 30} + {1 \over \sqrt{5}} )  ]^2
> $
>
> Finally this yields:
>
> $
>  {1\over 2}(1-p_\Delta)^{T_0-1} \ge {1\over 2}(1-e^{-a m})^{C_0m^2\ln m -1} \ge C_2
> $
>
> where $C_2$ is defined as
>
> $
> C_2 = \min_{m \ge 5} \Big( {1\over 2}(1-e^{-a m})^{C_0m^2\ln m -1} \Big)
> $
>
> We have that $C_2$ is non null since
>
> $\lim_{m \to \infty} {1\over 2}(1-e^{-a m})^{C_0m^2\ln m -1} = {1 \over 2} \ne 0$
>
> and one may readily check that $C_2$ is indeed a universal constant.
>
> Q2 : We proved line 504 that $P(\bar{\mathcal{A}}) > 1/2$. Then we use the fact that
>
> $P(\mathcal{B}_t) \ge P(\mathcal{B}_t , \bar{\mathcal{A}} ) = P(\mathcal{B}_t|\bar{\mathcal{A}} )P(\bar{\mathcal{A}}) \ge {1 \over 2} P(\mathcal{B}_t|\bar{\mathcal{A}} )$.
>
> Q3 : $C_3$ depends on $C_2$ and in fact we can simply choose $C_3 = C_2$ because $(1- { (2\Delta)^m \over m!})^{T-T_0} \ge (1- { (2\Delta)^m \over m!})^{T}$

---

> > ### Comment · Reviewer_dwfb · 2021-09-02
> > **Raising my score**
> >
> > Dear Authors,
> >
> > Thank you for your convincing and detailed rebuttals. I have increased my score to a 7 to reflect my opinion that the paper is interesting and will be strengthened by the edits outlined in the rebuttal.

---

### Official Review · Reviewer_nhUQ · 2021-07-13

**Rating:** 6
**Confidence:** 3

**Summary:**

This work considers combinatorial semi-bandit problems, and theoretically investigates the sub-obtimality of Thompson sampling in the high-dimensional feature space and empirically shows the poor performance of Thompson sampling compared to other existing approaches such as ESCB and CUCB algorithms.

**Limitations And Societal Impact:**

As mentioned in main review, this submission reports only weakness of Thompson sampling approach in combinatorial bandit in terms of both theoretical and empirical results. It can be a complete piece of work to suggest  TS-type algorithm with sub-linear regrets by handling its greedy property in high dimension.

**Main Review:**

Originality:
This work provides novel lower regret bounds for Thompson sampling to show that the minimax regret of Thompson sampling scales almost linearly when high dimensional feature is considered. Also it gives interesting examples for linear and non-linear combinatorial bandits, showing that regret of Thomason sampling scales exponentially in dimension.

Quality:
Theoretical results sound well-grounded and numerical experiments seem reasonably fair and supportive to this theoretical outcome, which makes a complete piece.

Clarity:
It was hard to read this submission since theorem and proofs are not clearly separated and it does not provide enough technical details in the proof. Forced exploration part seems redundant in a complete piece of work.

Significance:
This submission only proves the sub-optimality of Thompson sampling approach in combinatorial bandit. Though it left a new challenging open problem “how to design better Thompson-sampling type algorithm to deal with high dimensional features”, the significance of this work is weak without suggesting TS-type algorithm with sub-linear regrets.


**Time Spent Reviewing:**

4

---

> ### Author Response · Authors · 2021-08-09
> **Reviewer response**
>
> Dear reviewer,
> while, indeed we do not suggest new Thompson Sampling algorithms that alleviate the problem, we highlight that there exists other known (optimistic) algorithms that do not exhibit this problem at all. Therefore, a practitioner can simply chose those algorithms instead of Thompson sampling.  Most existing literature on bandits seems to operate under the assumption that Thompson sampling approaches are almost always superior to other approaches, and our work shows that this is not always the case.

---

### Official Review · Reviewer_qacF · 2021-07-19

**Rating:** 8
**Confidence:** 4

**Summary:**

This paper analyzes Thompson Sampling in combinatorial bandits problems and proves two lower bounds that show that in these problems, Thompson Sampling can scale exponentially in the dimension of the problem. In particular, they first construct an instance on which the regret of Thompson Sampling satisfies an instance dependent lower bound that scales exponentially with the dimension. They then show that in a "min-max" sense that the regret of TS on this instance can scale almost linearly unlike other optimism-based methods. Furthermore, they show that this sub-optimality is not alleviated by forcing exploration. They conclude with numerical results that verify their findings.



**Ethics Review Area:**

["I don’t know"]

**Limitations And Societal Impact:**

The point of this paper is to demonstrate the limitations of a popular algorithm in contextual bandit problems. I think the authors should be careful (in e.g., the title) to highlight that they show the limitations of thompson sampling in the combinatorial bandits problems and not more generally.

**Main Review:**

This paper is well written and represents a nice contribution showing that Thompson Sampling in combinatorial bandits is suboptimal with respect to the dimension of the problem. As the authors note, regret bounds for Thompson Sampling often show that it has worse dimension-dependence in problems than optimism based methods, but this is often assumed to be an artifact of the proof. This paper shows that--- in the case of combinatorial bandits---  this is not true and in fact this suboptimal scaling is unavoidable (even if you force exploration). This is a nice and surprising finding which expands on our growing understanding of Thompson Sampling.


Detailed Questions:

1. In linear bandits, Thompson Sampling is not yet known to achieve the optimal rate (compared to e.g., optimism based methods), and generally the variance of the posterior is blown up by a constant $\sqrt{d}$ in order to prove regret bounds (see e.g., [1], [2]). While--- to the best of my knowledge--- this factor has not been shown to be necessary, would scaling the variance similarly solve this scaling problem in combinatorial bandits? This seems to be different than forced exploration analyzed in Section 3.4.

2. How much of this suboptimality comes from a bad modeling decision? In the case of the first instance with two disjoint paths, if one reduced the problem to a two-arm MAB problem and used TS, the suboptimality would disapear. It seems to me like the modeling choice is overcomplicated for this problem and that this is the source of suboptimality while optimism based methods don't have this problem since they are more `black-box'. Is this one way to interpret the  results? The results in this paper are still interesting and relevant because they show the importance of choosing good models in thompson sampling, and the algorithm itself may not be suboptimal, but extremely reliant on simple models.

Minor:

1. There is a broken link to the appendix on line 134.


**Time Spent Reviewing:**

4 hours

---

> ### Author Response · Authors · 2021-08-09
> **Response to reviewer**
>
> Dear reviewer,
> 1. Indeed, artificially inflating the variance of Thompson samples would be another way of forcing the algorithm to be less greedy and increase exploration. We did not analyze this in details, and it certainly seems like an interesting open problem. However, our intuition is that if one were to increase the variance of Thompson samples by a factor of $\sqrt{d}$, the regret of this version of Thompson sampling would increase by a multiplicative factor that depends on $d$, so that this version of Thompson sampling would, once again, not exhibit the same regret scaling as optimistic algorithms.
> 2. We agree with your comment that modelling choices seem to have an impact on the performance of Thompson sampling. In fact the "modelling choices" can be summarized by the prior distribution chosen (here it is the uniform distribution). Our work shows that a badly chosen prior can have a catastrophic impact for the performance of Thompson.

---

### Decision · Program_Chairs · 2021-09-27

**Decision:**

Accept (Poster)

**Comment:**

The paper has received mixed reviews in the first round, but the author response has successfully addressed the concerns of the reviewers. After some internal discussion, the reviewers all agreed that the paper offers a strong and interesting contribution and should thus be accepted for publication at the conference.